# RAEE: A Robust Retrieval-Augmented Early Exit Framework for Efficient Inference

**Lianming Huang**[1*], **Shangyu Wu**[2*†], **Yufei Cui**[3], **Ying Xiong**[2], **Haibo Hu**[1],
**Xue Liu**[2,3], **Tei-Wei Kuo**[4], **Nan Guan**[1], **Chun Jason Xue**[2]

[1] City University of Hong Kong
[2] Mohamed bin Zayed University of Artificial Intelligence (MBZUAI)
[3] MILA, McGill University
[4] National Taiwan University

## Abstract

Deploying large language model inference remains challenging due to their high computational overhead. Early exit optimizes model inference by adaptively reducing the number of inference layers. Current methods typically train internal classifiers or use heuristic methods to determine the exit layer. However, those methods either introduce significant training overheads or lead to performance degradation. To address these limitations, this paper proposes RAEE, a robust **R**etrieval-**A**ugmented **E**arly **E**xit framework that not only enables early exit but also enhances model performance through corrective exit information at intermediate layers. This paper first demonstrates that the early exit problem can be effectively modeled as a distribution prediction problem, in which the distribution can be further approximated through the exit information of similar data. Subsequently, this paper introduces the process of collecting exit information of correct predictions and the steps to construct the retrieval database. Finally, leveraging the pre-constructed retrieval database, RAEE utilizes the exit information from retrieved similar data to guide the backbone model's exit. Experimental results demonstrate that RAEE can not only accelerate inference while achieving robust performance across eight downstream tasks.

## 1 Introduction

Large language models (LLMs) have been widely used in various application scenarios due to their excellent performance (Thoppilan et al., 2022; Touvron et al., 2023; Scao et al., 2022). However, the model inference efficiency still poses a great challenge due to high computational overhead and memory requirements (Dao et al., 2022; Liu et al., 2023). Early exit has emerged as an advanced model pruning method, which dynamically terminates the LLMs inference when a certain confidence criterion is met, thereby reducing latency and memory consumption (Valicenti et al., 2023; Ma et al., 2023).

Existing early exit frameworks (Liu et al., 2020; Zhu, 2021; Xin et al., 2020; Fan et al., 2024) can be categorized into three types: training-based, semi-training-based, and training-free approaches. *Training-based methods* (Zhu, 2021; Zhou et al., 2020; Zhu et al., 2023; Bae et al., 2023; Schuster et al., 2022) jointly optimize internal classifiers and the backbone model, incurring substantial training overhead. *Semi-training-based methods* (Fan et al., 2024) freeze the backbone model and only train the lightweight classifiers, but they often heavily rely on manual feature engineering and may not generalize well. *Training-free methods* (Sun et al., 2022) typically use heuristic exit criteria that lack adaptability and often lead to performance degradation compared to the full model. Notably, most early exit frameworks trade off accuracy for speed (Fan et al., 2024; Sun et al., 2022; Schuster et al., 2022; Bae et al., 2023).

---

[*] Authors contributed equally to this research.
[†] Corresponding author.

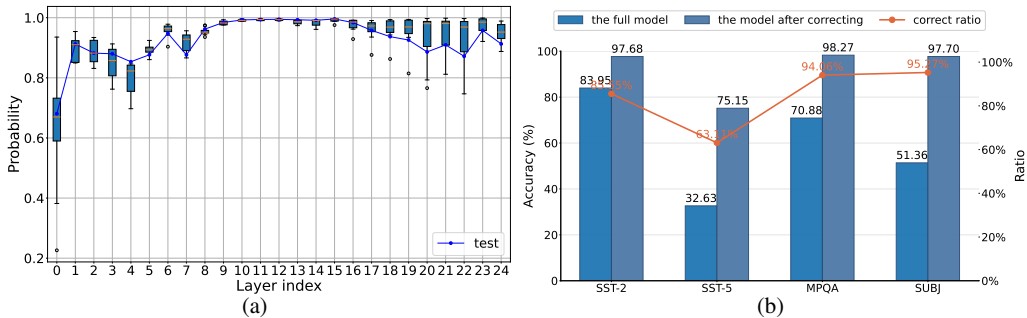

Figure 1: (a) Correct prediction probabilities for a test sample and its top-8 nearest neighbors from the SST-2 dataset. (b) The accuracy of the full model and the model after correcting from the intermediate layers, and the correct ratio. All results are collected with the backbone model RoBERTa-Large.

Different from existing early exit frameworks, we observe that early exit can not only accelerate the inference, but also act as a corrective mechanism when the full model makes the wrong predictions. Furthermore, we also observe that the exit behaviors are highly consistent across semantically similar data. These insights challenge the conventional trade-off scheme of early exit frameworks and also suggest that the key to an ideal early exit strategy lies in adaptively selecting the exit layer based on the behavior of similar instances.

Based on those observations, we propose RAEE, a **R**etrieval-**A**ugmented **E**arly **E**xit framework that improves both efficiency and accuracy without training classifiers. RAEE leverages the insights where similar inputs have similar exit behaviors to determine the exit layer. Specifically, we construct a retrieval database from external data by recording exit layers where correct predictions occur. During the inference, RAEE retrieves the exit information of the top-k similar examples and aggregates the results to determine the final exit layer.

We conduct comprehensive experiments to evaluate the proposed RAEE and various comparison methods on eight downstream tasks. Experimental results demonstrate that RAEE can accelerate the model inference while achieving robust model performance. Codes are available at [1].

The **main contributions** of this paper are:

- We model the early exit problem as a distribution prediction problem and demonstrate that the exit information of similar data can approximate the exit distribution;

- We propose RAEE, a robust retrieval-augmented early exit framework, which leverages an external database to guide the early exit;

- Experimental results show that the proposed RAEE can not only accelerate the model inference but also significantly improve the model accuracy, even sometimes surpassing the full model, which outperforms traditional early exit frameworks.

## 2 MOTIVATIONS

The primary goal of this work is to develop a robust and efficient early exit framework without training any classifiers or model parameters. To this end, we explore a retrieval-based diagram, which offers a simple yet effective way to augment the early exit framework during the inference stage. In this section, we begin with the definition of the early exit problem, then present key observations that motivate our approaches.

---

[1] https://github.com/HugeRaabbit/RAEE

## 2.1 PROBLEM STATEMENT

Formally, the early exit problem can be defined as follows: Given a backbone model $\mathcal{M}$ with $m$ layers and an input $x$, an early exit framework aims to design an exit function or classifier $l = f(x)$ to determine whether to exit at the layer $l$. The final prediction $y$ is then transformed from the intermediate output states $h_l$ of the $l$-th layer. And the final prediction probability can be formulated as $P(y \mid x) = P(y \mid h_{f(x)})$, where $f(x)$ is trained or built on the external data $\mathcal{D}$.

## 2.2 OBSERVATION 1: EARLY EXIT AS A CORRECTIVE MECHANISM

Conventional early exit frameworks are primarily designed to accelerate the inference at the expense of model accuracy. However, they may overlook the great potential of early exit as a corrective mechanism. This perspective indicates that *intermediate layers can sometimes make better predictions than the output of the final layer*. Therefore, we argue that early exit should be considered not only as an acceleration technique, but more as a dynamic mechanism to correct the model's output for each individual input.

To validate the above claims, we conducted analysis experiments using RoBERTa-large on the SST-2 datasets in Figure 1. Figure 1 (a) shows an example of the probabilities of predicting the correct answers of a testing sample and their top-8 nearest neighbor samples over different layers. As shown in Figure 1 (a), the model can already make correct predictions within the intermediate layers (layers 10-15), rather than performing all computations to obtain the final predictions. This indicates that if the model exits the inference from the intermediate layers, it may not degrade the overall performance.

Furthermore, we collect the correct prediction ratios of exiting from intermediate layers when the final predictions are wrong in Figure 1 (b). When the model makes the correct predictions, the way of exiting from the intermediate layers can achieve the same accuracy. This is because, in the worst case, the model can still obtain the correct predictions at the final layer even if the model cannot predict correctly in any intermediate layers. Figure 1 (b) shows that when the full model makes wrong predictions, 90.66% on average of those wrong predictions can be corrected by the outputs of intermediate layers if we can exit properly.

## 2.3 OBSERVATION 2: CONSISTENT EXIT BEHAVIORS OF SIMILAR DATA

Building on the corrective potential of early exit, a critical question arises: *how can we reliably identify the optimal exit layer for a given input?* We hypothesize that inputs with similar semantic content should exhibit similar optimal exit behaviors.

To validate this hypothesis, we build a retrieval database based on the exit behaviors of training data and collect the exit information of the top-8 nearest neighbors. As shown in Figure 1 (a), we observe that not only does the test sample show higher confidence at intermediate layers, but its top-8 nearest neighbors display a remarkably consistent pattern. Their probabilities of correct predictions are almost the same high at these intermediate layers (layers 10-15) compared to the final layer. This consistency suggests a great opportunity to leverage the top-k nearest neighbors' exit behaviors to approximate the given input's exit behavior.

The above observations motivate us to propose a retrieval-augmented early exit framework, which can not only accelerate the model inference by exiting from the intermediate layers but also further correct the model's wrong predictions, thus improving model accuracy.

## 3 METHODOLOGY

This section presents the retrieval-augmented early exit framework in detail. First, this paper outlines the process of collecting exit features and constructing the retrieval database to facilitate early exit. Then, this paper introduces the retrieval-augmented early exit framework, denoted as **RAEE**.

## 3.1 COLLECTING THE EXIT FEATURES AND BUILDING THE RETRIEVAL DATABASE

This paper uses the collected exit features as the keys and values within the retrieval database. To avoid introducing too much retrieving overheads, this paper only retrieves once at the beginning of

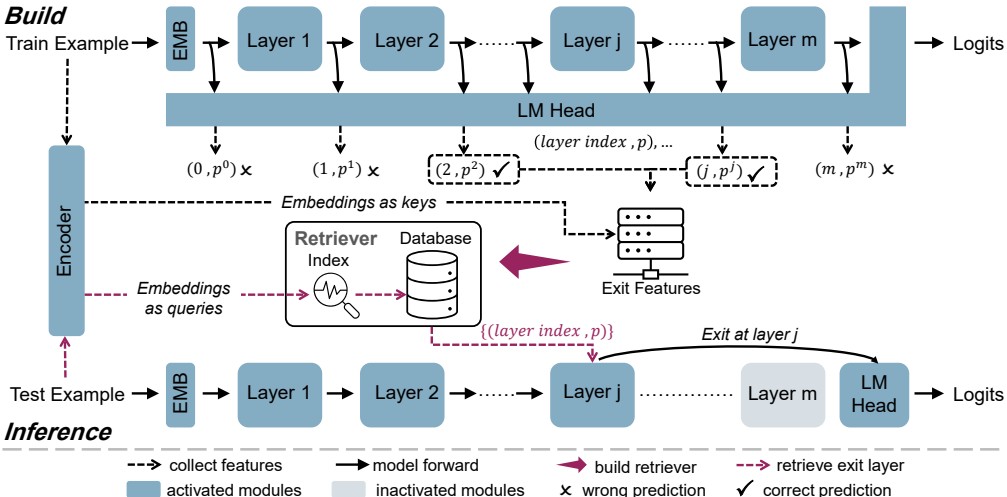

Figure 2: The overview of the retrieval-augmented early exit framework. During the build phase, a retrieval database is constructed from the collected exit features, including layer indexes and their corresponding probabilities for correct predictions. During the inference phase, the framework retrieves similar data's exit information based on data embeddings to guide the model in selecting the optimal exit layer.

the backbone model. Consider the training data $\mathcal{D} = \{(x_1^{train}, y_1^{train}), \ldots, (x_{|\mathcal{D}|}^{train}, y_{|\mathcal{D}|}^{train})\}$ and a backbone model $\mathcal{M}$ with $m$ layers $\{\mathcal{L}_1, \ldots, \mathcal{L}_m\}$. In this context, as shown in the top part of Figure 2, the keys $\mathcal{K}$ are input embeddings of the training data, which can be obtained from an extra encoder model $\mathcal{E}$, such as BERT (Devlin et al., 2019), or the outputs of embedding layers in the backbone model $\mathcal{M}_{emb}$,

$$\mathcal{K} = \{e_i\}_{i=1}^{|\mathcal{D}|} = \{\mathcal{E}(x_i^{train})\}_{i=1}^{|\mathcal{D}|}. \tag{1}$$

For the values, this paper collects a set of possible exit layers $l_i$ and corresponding probabilities $p_i$ for each embedding $e_i$, i.e., $v_i = \{(l_i^j, p_i^j)\}_{j=1}^{m_i}$, where $m_i$ indicates the number of possible exit layers for the embedding $e_i$. The layer $l$ chosen as the exit layer is determined by whether the outputs of this layer $h_l$ can be used to make the right predictions $\hat{y}$ compared to the training labels $y^{train}$. Then, the values $\mathcal{V}$ are all sets of possible exit layers,

$$\mathcal{V} = \{v_i\}_{i=1}^{|D|} = \left\{ \{(l_i^j, p_i^j)\}_{j=1}^{m_i} \right\}_{i=1}^{|D|}. \tag{2}$$

We follow the same dataset splitting used in the LM-BFF (Gao et al., 2021), *the collecting process requires no parameters of the model to update, only model inference is performed.*

After collecting keys and values for the retrieval databases, this paper uses state-of-the-art approximate nearest neighbor search indexing, such as FAISS (Johnson et al., 2019), and efficient key-value stores to build the retrieval database. More details can be found in Algorithm C.1 of Appendix C.

## 3.2 THE RETRIEVAL-AUGMENTED EARLY EXIT FRAMEWORK

In this section, this paper then presents a retrieval-augmented early exit framework named RAEE to optimize the model inference. RAEE regards the exit layer as a random variable $z$, taking values in the set of $\{1, \ldots, m\}$, where $m$ is the total number of layers in the backbone model $\mathcal{M}$. The probability mass function $P(z = l)$ represents the probability of the case that the backbone model exits at the layer $l$. With the gold label, we can observe that the random variable $z$ follows an unknown discrete distribution $F$. Then, this paper shows how to leverage the retrieval database to approximate the distribution $F$.

Given an input $x$, RAEE first retrieves top-$k$ nearest neighbors $\{v_1, \ldots, v_k\}$, where each neighbor $v_i$ has $m_i$ possible exit layers. Naturally, we can approximate the distribution $F$ by estimating the probability function $P(z = l)$,

Table 1: Model performance of different methods on eight downstream tasks. 'RB-L', 'EB-L', and 'T5-L' refer to RoBERTa-Large, ElasticBERT-Large, and T5-Large, respectively.

| Methods | SST-2 | SST-5 | MR | CR | MPQA | SUBJ | TREC | CoLA | Avg |
|---|---|---|---|---|---|---|---|---|---|
| RB-L | 83.60 | 34.98 | 80.80 | 79.55 | 67.60 | 51.45 | 32.40 | 2.03 | 54.05 |
| EB-L | 51.15 | 22.35 | 49.25 | 48.65 | 48.05 | 48.85 | 17.60 | 0.11 | 35.75 |
| T5-L | 49.31 | 23.12 | 50.40 | 50.90 | 45.40 | 52.75 | 27.60 | -4.64 | 36.86 |
| Llama-3-8B | 62.84 | 26.06 | 59.65 | 72.90 | 51.75 | 52.80 | 8.40 | 0.00 | 41.80 |
| Gemma-7B | 49.08 | 28.64 | 50.05 | 50.10 | 50.00 | 48.05 | 14.40 | -0.79 | 36.19 |
| *Backbone: RoBERTa-Large, ElasticBERT-Large* | | | | | | | | | |
| HashEE (EB-L) | 49.08 | 14.16 | 49.95 | 50.05 | 50.00 | 50.00 | 27.00 | 0.00 | 36.28 |
| DeeBERT (RB-L) | 52.29 | 18.05 | 50.60 | 50.00 | 75.95 | 80.85 | 16.20 | 0.00 | 42.99 |
| AdaInfer (RB-L) | 50.92 | 24.48 | 50.00 | 50.00 | 60.90 | 50.85 | 22.60 | -1.62 | 38.52 |
| RAEE (RB-L) | 84.63 | 33.57 | 81.55 | 68.05 | 78.55 | 84.05 | 62.40 | 14.48 | **63.41** |
| *Backbone: T5-Large* | | | | | | | | | |
| CALM (T5-L) | 51.72 | 23.17 | 49.25 | 50.55 | 49.80 | 49.90 | 18.00 | 0.00 | 36.55 |
| AdaInfer (T5-L) | 50.11 | 28.14 | 50.35 | 49.80 | 46.30 | 49.95 | 26.00 | 5.22 | 38.23 |
| RAEE (T5-L) | 52.98 | 26.56 | 50.80 | 51.60 | 55.65 | 49.90 | 39.80 | 12.20 | **42.44** |
| *Backbone: Llama-3-8B* | | | | | | | | | |
| SLEB (Llama) | 54.01 | 21.09 | 51.10 | 49.45 | 55.65 | 49.95 | 14.00 | 0.92 | 37.02 |
| AdaInfer (Llama) | 53.21 | 18.05 | 53.50 | 50.00 | 49.95 | 47.55 | 16.20 | 0.00 | 36.06 |
| RAEE (Llama) | 73.05 | 35.25 | 66.45 | 57.95 | 75.05 | 90.05 | 51.80 | 9.55 | **57.39** |
| *Backbone: Gemma-7B* | | | | | | | | | |
| SLEB (Gemma) | 50.69 | 19.82 | 49.95 | 49.95 | 50.00 | 52.10 | 12.80 | 0.00 | 35.66 |
| AdaInfer (Gemma) | 50.92 | 12.62 | 50.00 | 50.00 | 50.00 | 50.60 | 22.60 | 0.00 | 35.84 |
| RAEE (Gemma) | 73.17 | 32.40 | 66.75 | 56.75 | 75.60 | 90.15 | 40.00 | 10.46 | **55.66** |

$$P(z = l \mid x) = \sum_{i=1}^{k} P(v_i \mid x) \cdot S_i,$$
$$S_i = \sum_{j}^{m_i} \mathbb{1}\Big(any(l_i^j = l \,\&\&\, p_i^j \geq \tau)\Big) \cdot p_i^j. \tag{3}$$

Where $\mathbb{1}$ is the indicator function that returns 1 if the condition is true and 0 otherwise, $any(\cdot)$ is the function that returns true if one condition is true and false otherwise, $\tau$ is the threshold for filtering the layers, the inner loop only count once since there is at most one possible exit layer of neighbor $i$ that is equalt to $l$. Since different neighbors should have different contributions to the probability function $P(z = l)$, RAEE uses the reciprocal of the scaled distance between each neighbor and the query to estimate the contribution,

$$P(v_i \mid x) = \frac{\min\left(\{distance(v_j, x)\}_{j=1}^{k}\right)}{distance(v_i, x)}. \tag{4}$$

Then, RAEE designs a function $f(x)$ to determine the exit layer, which selects the layer that maximizes the probability function $P(z = l)$,

$$f(x) = \arg\max_{l} P(z = l \mid x). \tag{5}$$

Notably, when multiple exit layers have the same maximal probability, RAEE selects the earliest one.

The bottom part of Figure 2 shows the inference workflow of RAEE. Specifically, RAEE first feeds the inputs into the backbone model for the label predictions and the same encoder used in the building process for the query embeddings. Then, the retriever in RAEE retrieves the top-$k$ nearest neighbors in the retrieval databases based on the query embeddings. After obtaining all possible exit layers of $k$ nearest neighbors, RAEE computes the exit layers based on the Equations 3-5. Finally, RAEE stops the forwarding at the calculated exit layer and passes the intermediate outputs of the exit layer to the final prediction layer, e.g., LM Head in language models, to obtain the final predictions. This

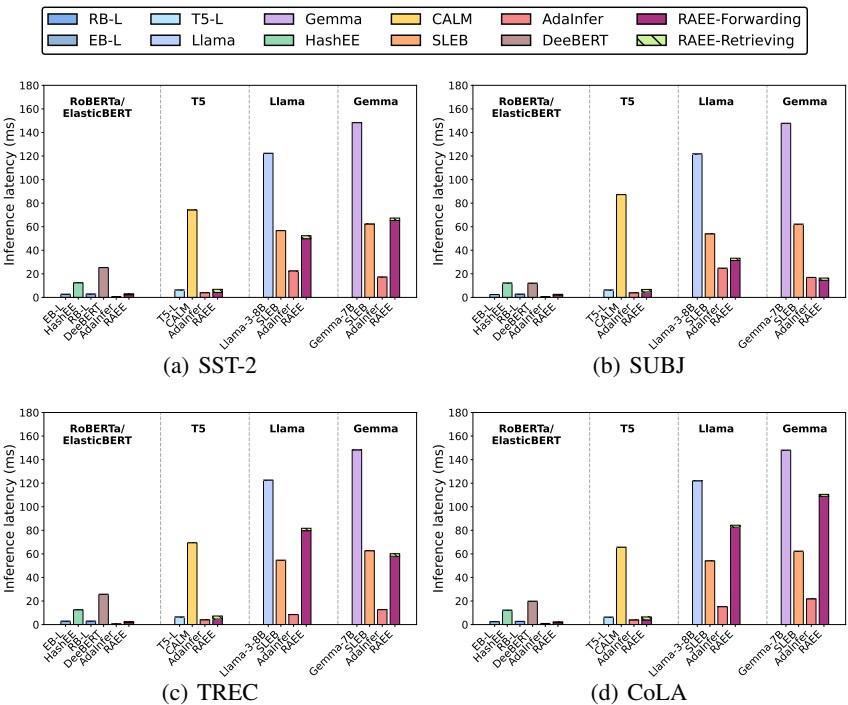

Figure 3: Inference latency of RAEE compared with various methods on selected downstream tasks. The backbone models used in the comparisons include RoBERTa-Large/ElasticBERT-Large, T5-Large, Llama-3-8B, and Gemma-7B.

is implemented based on the Transformer library, passing the exit layer as a parameter into the 'forward()' function and stopping the inner iteration based on the exit layer. More details can be found in Algorithm C.2 of Appendix C.

## 4 EXPERIMENTS

In this section, we first introduce the dataset and the experimental setup. Then, we presented the main results of different methods on eight downstream tasks. We also conducted analysis experiments and ablation studies to show the impact of these factors on RAEE performance.

### 4.1 DATASET AND EXPERIMENTAL SETUP

**Datasets** We conducted comprehensive experiments across eight downstream tasks from GLUE benchmark (Wang et al., 2019), covering sentiment analysis, opinion polarity analysis, grammatical judgment, natural language inference, paraphrasing, etc.

**Experimental Setup** The proposed RAEE was implemented using the PyTorch framework and Transformer. We evaluated methods based on the backbone models RoBERTa-Large (Liu et al., 2019) and T5-Large (Raffel et al., 2020) on one NVIDIA GeForce RTX 4090 with 24GB GPU memory, while Llama-3-8B (Dubey et al., 2024) and Gemma-7B (Mesnard et al., 2024) on one NVIDIA A100 GPU with 40GB GPU memory. The experiments were conducted in two settings: training-free settings for methods that require no parameter updates, and fine-tuning settings for methods that require searching for the optimal parameter of exit classifiers. The evaluation metric is accuracy, except for the Matthew correlation coefficient for the CoLA task. The number of retrieved nearest neighbors of RAEE is set to 12 in the experiments. The threshold $\tau$ of RAEE is set to 0.9.

To validate the effectiveness, we compared RAEE with three types of methods. **Pretrained Models:** 1) RoBERTa-Large (Liu et al., 2019), a state-of-the-art encoder model, where the prompt-based version (Gao et al., 2021) is used; 2) ElasticBERT (Liu et al., 2022), a pre-trained multi-exit

Table 2: Model performance and inference latency of RAEE and RAEE with only correctly predicted examples in the retrieval database. The backbone model is Llama-3-8B.

| Models | SST-2 | SST-5 | MR | CR | MPQA | SubJ | TREC | CoLA | Avg |
|---|---|---|---|---|---|---|---|---|---|
| *Performance* ↑ | | | | | | | | | |
| Llama | 62.84 | 26.06 | 59.65 | 72.90 | 51.75 | 52.80 | 8.40 | 0.00 | 41.80 |
| RAEE w/o | 60.55 | 24.52 | 57.30 | 53.55 | 56.65 | 81.70 | 20.80 | 0.00 | 44.38 |
| RAEE | 73.05 | 35.25 | 66.45 | 57.95 | 75.05 | 90.05 | 51.80 | 9.55 | **57.39** |
| *Latency (ms)* ↓ | | | | | | | | | |
| Llama | 122.27 | 122.13 | 122.03 | 121.78 | 121.82 | 121.70 | 122.52 | 122.06 | 122.04 |
| RAEE w/o | 37.65 | 115.26 | 37.98 | 31.69 | 54.08 | 34.59 | 112.03 | 91.34 | 64.33 |
| RAEE | 52.33 | 65.47 | 53.83 | 34.65 | 55.02 | 33.25 | 81.74 | 84.34 | **57.58** |

transformer model, where the large version is used in this paper; 3) T5-Large (Raffel et al., 2020), a versatile transformer-based model for various NLP tasks; 4) Llama-3-8B (Dubey et al., 2024), a pre-trained model with strength in specific language scenarios; 5) Gemma-7B (Mesnard et al., 2024), a model with the potential for outstanding performance in specific settings. **Training-Free Methods:** 1) HashEE (Sun et al., 2022), a hash-based early exit approach with ElasticBERT-Large as its backbone model; 2) CALM (Schuster et al., 2022), a classical entropy-thresholding-based early exit method with T5-Large as its backbone model, where the training-free setting is applied; 3) SLEB (Song et al., 2024), a method that eliminates redundant transformer blocks. **Semi-Training Methods:** 1) AdaInfer (Fan et al., 2024), an SVM-based early exiting method with our reproduced version; 2) DeeBERT (Xin et al., 2020), a classical entropy-thresholding-based early exiting method with RoBERTa-Large as its backbone model; The templates are listed in the Appendix B. More details about the experimental setup can be found in Appendix E.

## 4.2 MAIN RESULTS

Table 1 presents the main results, comparing the performance of RAEE against different types of methods across eight downstream tasks. Experimental results show that RAEE achieves the highest average performance of 63.41 with RoBERTa-Large. RAEE with Gemma-7B achieves the maximal improvement over baseline models, while with RoBERTa-Large, it boosts performance from 36.28 to 63.41 over comparison methods. Across eight downstream tasks, RAEE consistently improves the model performance compared to current state-of-the-art early exit frameworks.

Figure 3 shows the inference latency of RAEE and comparisons on eight downstream tasks. We show the inference latency on the selected four tasks, which covers different task types. For million-level backbone models, such as RoBERTa-Large, ElasticBERT-Large, T5-Large, RAEE can achieve comparable inference efficiency. Since the inference speeds of those backbone models are already fast enough, adding too many components for early exit would degrade the inference efficiency like HashEE and DeeBERT. However, for billion-level backbone models, the acceleration of RAEE is significant. This benefits from the effectively predicted early exit layers. Although Adainfer with the backbone Llama-3-8B and Gemma-7B is much faster than RAEE, it only achieves a comparable performance of backbone models. RAEE **significantly improves the performance of those backbone models** and also reduces the inference latency by nearly half.

## 4.3 REASONS FOR SIGNIFICANT PERFORMANCE IMPROVEMENT

The main results demonstrate that the proposed RAEE can significantly outperform backbone models, which is not an intuitive result compared to previous early exit methods. The reasons for this lie in that **the collected exit information guides the RAEE as an error corrector**. This means RAEE can leverage exit information from cases correctly predicted by intermediate layers but missed by the full-backbone models without early exit.

To further validate the above claims, we performed an analysis using the retrieval database, which contains exit information only from the data correctly predicted by the full Llama-3-8B without early exit. As shown in Table 2, RAEE w/o refers to the one built on only correctly predicted examples. It can be seen that RAEE w/o achieves comparable performance to baselines while accelerating the

Table 3: The impact of retrieval number $k$ on the distribution approximation.

| $k$ | SST-2 | SST-5 | MR | CR | MPQA | SUBJ | TREC | CoLA | Avg |
|---|---|---|---|---|---|---|---|---|---|
| 2 | 79.82 | 32.99 | 76.40 | 68.40 | 77.55 | 81.75 | 61.00 | 7.89 | 60.73 |
| 4 | 82.22 | 33.17 | 79.20 | 68.05 | 78.65 | 83.60 | 63.60 | 14.74 | 62.90 |
| 8 | 84.52 | 34.12 | 80.70 | 68.05 | 78.90 | 84.20 | 63.20 | 12.19 | 63.24 |
| 12 | 84.63 | 33.57 | 81.55 | 68.05 | 78.55 | 84.05 | 62.40 | 14.48 | **63.41** |
| 16 | 84.98 | 32.17 | 82.05 | 69.00 | 77.90 | 84.00 | 62.60 | 13.00 | 63.21 |
| 20 | 85.44 | 32.26 | 81.80 | 68.95 | 78.05 | 83.50 | 62.40 | 12.40 | 63.10 |

Table 4: The impact of retrieval database size on the distribution approximation. The percentage refers to the amount of training data that is used to build the retrieval database.

| Database Size | SST-2 | SST-5 | MR | CR | MPQA | SUBJ | TREC | CoLA | Avg |
|---|---|---|---|---|---|---|---|---|---|
| 20% | 84.63 | 31.76 | 82.80 | 65.85 | 75.70 | 81.00 | 58.80 | 8.77 | 61.16 |
| 50% | 83.37 | 32.85 | 80.65 | 66.95 | 77.90 | 83.60 | 61.20 | 11.73 | 62.28 |
| 100% | 84.63 | 33.57 | 81.55 | 68.05 | 78.55 | 84.05 | 62.40 | 14.48 | **63.41** |

inference process. This is because on test examples, for which the full backbone can correctly predict, RAEE w/o almost achieves the same correct predictions while exiting early. However, due to a lack of exit information on examples where the full backbone fails to predict, RAEE w/o also fails to predict on the test data where backbone models fail. Therefore, when providing the exit information based on examples where backbone models fail to predict but intermediate outputs succeed in predicting, RAEE can make correct predictions and exit earlier.

## 4.4 ABLATION STUDY

**Impact of Top-$k$:** Table 3 shows the effect of varying retrieval numbers on distribution approximation. With RoBERTa-Large, RAEE improves from 60.73 to 63.41 as $k$ increases, indicating that more retrieved exit information enhances approximation. However, beyond $k = 12$, performance drops to 63.10, likely due to noise from less relevant retrievals. This suggests that a limited amount of relevant exit data suffices for accurate approximation.

**Retrieval Database Size:** Table 4 shows the performance of RAEE with different sizes of retrieval databases. As the database size increases, the performance of RAEE with the backbone model RoBERTa-Large increases significantly from 61.16 to 63.41 on average. This demonstrates that collecting more data can improve the generalization of RAEE, thus approximating the exit distribution more accurately.

## 4.5 OUT-OF-DOMAIN PERFORMANCE

To investigate out-of-domain issues, we conducted experiments on summarization tasks, including CNN/DailyMail (See et al., 2017) and XSum (Narayan et al., 2018), using a retrieval database based on WikiText (Merity et al., 2017), as shown in Table 5. The results demonstrate that RAEE enhances performance and exits earlier despite the retrieval database being out of distribution. Detailed experimental settings are provided in Appendix F.

Table 5: Model performance (ROUGE-L) and layer usage on generation tasks. The backbone model is Llama-3-8B.

| Models | CNN/DailyMail | XSum |
|---|---|---|
| *Performance (ROUGE-L)* | | |
| Llama | 8.95 | 5.22 |
| RAEE Wiki | 14.01 | 7.15 |
| *Layers* | | |
| Llama | 32.00 | 32.00 |
| RAEE Wiki | 29.60 | 28.82 |

Table 6: The building cost of RAEE and different comparison methods. The results are collected with the backbone model RoBERTa-Large (ElasticBERT for HashEE).

| Model | SST-2 | SST-5 | MR | CR | MPQA | Subj | TREC | CoLA | Avg |
|---|---|---|---|---|---|---|---|---|---|
| Time Cost (seconds) | | | | | | | | | |
| RAEE | 91.75 | 111.80 | 112.09 | 27.43 | 110.26 | 103.99 | 71.19 | 108.59 | 92.14 |
| HashEE | 5.85 | 14.13 | 14.68 | 2.82 | 10.40 | 13.84 | 7.50 | 5.81 | 9.38 |
| AdaInfer | 91.85 | 124.78 | 129.11 | 41.77 | 122.96 | 120.52 | 66.43 | 114.19 | 101.45 |
| Storage Overheads (MB) | | | | | | | | | |
| RAEE (Index) | 3.4 | 3.7 | 3.8 | 2.0 | 3.8 | 3.6 | 3.1 | 3.7 | 3.4 |
| RAEE (DB) | 2.5 | 2.0 | 3.0 | 0.8 | 2.5 | 2.7 | 1.0 | 2.6 | 2.1 |

## 4.6 BUILDING OVERHEADS

We present the building overheads in Table 6, including database size, index size, and building time costs for different methods. Specifically, the building time cost for RAEE refers to the time required to build the retriever, while the building time cost for AdaInfer pertains to the time needed for training the classifier. In the case of HashEE, the building time cost corresponds to the time taken to build the hashing buckets. For RAEE, the average time required to build the retrieval database is under 2 minutes on a single NVIDIA GeForce RTX 4090, which is considered an acceptable overhead in comparison to the time involved in fine-tuning. The index and database sizes are relatively small and can be considered negligible in comparison to the size of the backbone model.

## 5 RELATED WORK

### 5.1 EARLY EXIT FRAMEWORK

Early exit inference is a popular pruning method to reduce computation overhead in text tasks. Most current works (Bae et al., 2023; Kong et al., 2022; Ji et al., 2023; Wolczyk et al., 2021; Hooper et al., 2023) introduce classifiers in each layer to determine whether the inference should continue. (Xin et al., 2020; Liu et al., 2020; Zhou et al., 2020; He et al., 2021; Bajpai & Hanawal, 2025b) train classifiers by aligning intermediate and final outputs, then apply early exit based on a threshold. (Liao et al., 2021) incorporates the past and future information to predict the early exit layer. Instead of training a neural network as classifiers, (Fan et al., 2024) only fit the machine learning classifiers on the extracted features for the early exit. (Zhu, 2021; Zhu et al., 2021; 2023; Zhang et al., 2023a) focus on designing novel loss functions for training a more robust classifier. (Li et al., 2021) incorporates sentence-level features as well as token-level features to predict the early exit. Different from those works, our method does not require training the classifier. (Sun et al., 2022) proposes a hash-based early exit method that uses the hashing functions to map tokens to exit layers. (Bajpai & Hanawal, 2024) proposes an online learning algorithm for BERT, dynamically setting exit points based on confidence thresholds. (Regol et al., 2023) proposes a jointly-learned framework for early exiting and inference in dynamic neural networks, integrating gating mechanisms and intermediate inference modules. (Balagansky & Gavrilov, 2022) introduces a deterministic Q-exit criterion and revising the model architecture. (Jazbec et al., 2024) further develops a theoretically grounded, risk-controlled exit rule that can wrap around existing dynamic models. Our method uses pre-built databases to predict exit layers, achieving better generalization and even outperforming the full backbone model. Other works (Jazbec et al., 2023; Li et al., 2023; Huang et al., 2018; Bajpai & Hanawal, 2025a; Meronen et al., 2024) focus on designing early exit frameworks for image classification tasks or multimodal tasks. They are optimizations in the different domains compared to our works.

### 5.2 RETRIEVAL-BASED AUGMENTATIONS

Retrieval-based augmentations (Li et al., 2022; Wang et al., 2023; Xiong et al., 2023; Cui et al., 2023; Wu et al., 2024a;b) have been widely used in various natural language processing (NLP) tasks and achieved remarkable performance. Current works mostly leverage external knowledge databases to augment generator models on various text-generation tasks, such as language modeling (Khandelwal

et al., 2020; Lewis et al., 2020; Borgeaud et al., 2022), question-answering (Guu et al., 2020; Izacard & Grave, 2021), machine translation (Khandelwal et al., 2021; Wang et al., 2022), dialogue system (Cheng et al., 2023). Those works focus only on improving generation quality, while our approach simultaneously accelerates inference and enhances model performance. (Zhang et al., 2023b) stores the generation trajectories of diffusion models that differ from our target models and retrieves similar trajectories to skip several intermediate sampling steps. These works are out of the scope of the research problems in this paper.

## 6 CONCLUSION

This paper models the early exit problem as a distribution prediction problem and observes that similar data's exit information can be used to approximate the distribution. Based on the observations, this paper proposes a retrieval-augmented early exit framework named RAEE. Experimental results show that RAEE can accelerate the model inference while significantly improving the model performance.

## 7 ACKNOWLEDGEMENT

This work was partially supported by Guangdong Provincial 1+1+1 Grant.

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

# A  THE USE OF LARGE LANGUAGE MODELS

We used a large language model mainly for language polishing (grammar, phrasing, and clarity) in our paper. All LLM suggestions were manually reviewed by the authors to ensure accuracy and consistency with our research objectives and writing style.

# B  TEMPLATES ON ALL TASKS

Table 7 provides an overview of the manual templates and selected label words used for each dataset with the backbone model RoBERTa-Large (Liu et al., 2019) in this paper. These templates and label words were created following LM-BFF (Gao et al., 2021).

Table 7: Templates and label words with the backbone model RoBERTa-Large.

| Task | Prompts | Label word |
|------|---------|-----------|
| SST-2 | [CLS] $x$ It was [MASK]. [SEP] | "0":"terrible", "1":"great" |
| SST-5 | [CLS] $x$ It was [MASK]. [SEP] | "0":"terrible","1": "bad", "2": "okay","3": "good","4": "great" |
| MR | [CLS] $x$ It was [MASK]. [SEP] | "0":"terrible", "1":"great" |
| CR | [CLS] $x$ It was [MASK]. [SEP] | "0":"terrible", "1":"great" |
| MPQA | [CLS] $x$ It was [MASK]. [SEP] | "0":"terrible", "1":"great" |
| SUBJ | [CLS] $x$ This is [MASK]. [SEP] | "0":"subjective", "1":"objective" |
| TREC | [CLS] [MASK] $x$ [SEP] | "0":"Description","1":"Entity","2":"Expression", "3":"Human","4":"Location","5":"Number" |
| CoLA | [CLS] $x$ It was [MASK]. [SEP] | "0":"incorrect", "1":"correct" |

Table 8 provides an overview of the manual templates and selected label words used for each dataset with the backbone model T5-Large (Raffel et al., 2020), Llama-3-8B (Dubey et al., 2024) and Gemma-7B (Mesnard et al., 2024) in this paper.

Table 8: Templates and label words with the backbone model T5-Large, Llama-3-8B and Gemma-7B.

| Task | Prompts | Label word |
|------|---------|-----------|
| SST-2 | What is the sentiment of the sentence $x$ ? Print negative or positive. The answer is | "0":"negative", "1":"positive" |
| SST-5 | What is the sentiment of the sentence $x$ '? Print terrible, bad, okay, good or great. The answer is | "0":"terrible","1": "bad", "2": "okay","3": "good", "4": "great" |
| MR | What is the sentiment of the sentence $x$ ? Print negative or positive. The answer is | "0":"negative", "1":"positive" |
| CR | What is the sentiment of the sentence $x$ ? Print negative or positive. The answer is | "0":"negative", "1":"positive" |
| MPQA | What is the sentiment of the sentence $x$ ? Print negative or positive. The answer is | "0":"negative", "1":"positive" |
| SUBJ | What is the subjectivity of the sentence $x$ ? Print subjective or objective. The answer is | "0":"subjective", "1":"objective" |
| TREC | Print the category for the sentence $x$ : description, entity, expression, person, location or quantity. The answer is | "0":"description","1":"entity", "2":"expression","3":"person", "4":"location","5":"quantity" |
| CoLA | Is the sentence $x$ grammatically acceptable? Print no or yes. The answer is | "0":"no", "1":"yes" |

---

**Algorithm C.1** Collect the exit features as keys and values for building the retrieval database.

---

**Input:** Given the training data available for model inference $\mathcal{D} = \{(x_1^{train}, y_1^{train}), \ldots, (x_{|\mathcal{D}|}^{train}, y_{|\mathcal{D}|}^{train})\}$,
  backbone model $\mathcal{M}$ with $m$ layers $\{\mathcal{L}_1, \ldots, \mathcal{L}_m\}$, encoder $\mathcal{E}$ (None value means no encoder is provided).
**Output:** Keys $\mathcal{K}$ and values $\mathcal{V}$.
  1: $\mathcal{K} = [], \mathcal{V} = []$
  2: **for** $i = 1, \ldots, |\mathcal{D}|$ **do**
  3:    $v_i = []$;
  4:    $h_0 = \mathcal{M}_{emb}(x_i^{train})$;
  5:    **for** j=1, ..., m **do**
  6:       $h_j = \mathcal{L}_j(h_{j-1})$;                       /* Compute intermediate outputs at layer $j$ */
  7:       $logits = \mathcal{M}_{lm\_head}(h_j)$;                        /* Predict from the layer $j$ */
  8:       $\hat{y} = \arg\max logits$;
  9:       $p_i^j = \max\{\text{softmax}(logits)\}$ ;
 10:       **if** $\hat{y}$ is equal to $y_i^{train}$ **then**
 11:          Add $(j, p_i^j)$ into $v_i$;                       /* Store the possible exit layer */
 12:       **end if**
 13:    **end for**
 14:    Add $v_i$ into $\mathcal{V}$;
 15:    **if** $\mathcal{E}$ is None **then**
 16:       Add $h_0$ into $\mathcal{K}$;               /* Store backbone embeddings when no encoder model */
 17:    **end if**
 18: **end for**
 19: **if** $\mathcal{E}$ is not None **then**
 20:    Add all $\mathcal{E}(x_i^{train})$ into $\mathcal{K}$;                       /* Store encoder embeddings */
 21: **end if**
 22: **return** $\mathcal{K}, \mathcal{V}$;

---

## C    DETAILED ALGORITHMS OF RAEE

Algorithm C.1 collects keys and values for building the retrieval database. For each sample in training data $\mathcal{D}$, the backbone model $\mathcal{M}$ is traversed layer by layer to compute the hidden state $h_j$ and corresponding $logits$. If a prediction $\hat{y}$ at a certain layer $j$ matches the sample's true label $y_i^{train}$, the exiting information, including the layer $j$ and the probability $p_i^j$, is added to the sample's value list (Line 2-12). When the encoder $\mathcal{E}$ is unavailable (Line 16), RAEE utilizes the hidden states from the backbone model $\mathcal{M}$ as embeddings for indexing. The specific layer from which the hidden states are extracted is treated as a hyperparameter that the user can define.

Algorithm C.2 performs model inference with retrieval-augmented early exiting. When the encoder $\mathcal{E}$ is unavailable (Line 5), RAEE utilizes the hidden states from the backbone model $\mathcal{M}$ as embeddings for querying. The specific layer from which the hidden states are extracted is treated as a hyperparameter that the user can define.

## D    EXIT LAYERS

Table 9 compares the average exit layers of the RAEE method against two other method types across eight downstream tasks. Experimental results show that the RAEE method can exit earlier, thus reducing computational overhead during model inference. This result also aligns with the expectations in the motivation example. This suggests that the RAEE method can accurately approximate the gold exit layer distribution by using the retrieval database. Although AdaInfer exits earlier than the RAEE method, it exhibits quite poor performance, as shown in Table 1. The reason may be that the collected features can only provide limited information for the SVM, thus resulting in unstable prediction performance.

## E    IMPLEMENTATION DETAILS

This section lists the implementation details.

---

**Algorithm C.2** Model inference with the synchronized retrieval-augmented early exit.

---

**Input:** Input $x$, backbone model $\mathcal{M}$ with $m$ layers $\{\mathcal{L}_1, \ldots, \mathcal{L}_m\}$, encoder $\mathcal{E}$, indexing $\mathcal{I}$, top-$k$, the exit layer determination function $f(\cdot)$.

**Output:** Final prediction $\hat{y}$.

1:  $h_0 = \mathcal{M}_{emb}(x)$;
2:  **if** $\mathcal{E}$ is not None **then**
3:      $e_{query} = \mathcal{E}(x)$;                                          /* Encode the inputs when the encoder is available */
4:  **else**
5:      $e_{query} = h_0$;                                          /* Use the embeddings of backbone model */
6:  **end if**
7:  $\{(v_i, dis_i)\}_{i=1}^k = \mathcal{I}(e_{query}, k)$;                                          /* Retrieve the possible exit layers */
8:  $l = f(\{(v_1, dis_1), \ldots, (v_k, dis_k)\})$;                                          /* Obtain the exit layer */
9:  **for** $i = 1, \ldots, l$ **do**
10:     $h_i = \mathcal{L}_i(h_{i-1})$;                                          /* Perform model inference with early exit */
11: **end for**
12: $logits = \mathcal{M}_{lm\_head}(h_l)$;                                          /* Predict based on the layer $h_l$ outputs */
13: $\hat{y} = \arg\max logits$;
14: **return** $\hat{y}$;

---

Table 9: Exit layers of RAEE and different types of methods on 8 downstream tasks. The sum of the number of layers in the encoder and the decoder counts the number of layers for T5-large (Raffel et al., 2020).

| Model | SST-2 | SST-5 | MR | CR | MPQA | Subj | TREC | CoLA | Avg |
|---|---|---|---|---|---|---|---|---|---|
| RB-L | 24 | 24 | 24 | 24 | 24 | 24 | 24 | 24 | 24 |
| EB-L | 24 | 24 | 24 | 24 | 24 | 24 | 24 | 24 | 24 |
| T5-L | 48 | 48 | 48 | 48 | 48 | 48 | 48 | 48 | 48 |
| Llama-3-8B | 32 | 32 | 32 | 32 | 32 | 32 | 32 | 32 | 32 |
| Gemma-7B | 28 | 28 | 28 | 28 | 28 | 28 | 28 | 28 | 28 |
| *Backbone: RoBERTa-Large, ElasticBERT-Large* | | | | | | | | | |
| DeeBERT | 22.95 | 24.00 | 23.33 | 8.98 | 15.90 | 10.36 | 24.00 | 18.31 | 18.48 |
| AdaInfer (RB-L) | 1.00 | 0.00 | 1.46 | 1.00 | 18.00 | 1.10 | 0.00 | 4.00 | 3.32 |
| *RAEE* (RB-L) | 18.55 | 13.93 | 18.71 | 15.35 | 17.20 | 13.59 | 12.82 | 12.48 | 15.33 |
| *Backbone: T5-L* | | | | | | | | | |
| AdaInfer (T5-L) | 6.34 | 0.00 | 7.72 | 0.00 | 1.00 | 1.00 | 0.00 | 1.00 | 2.13 |
| *RAEE* (T5-L) | 22.27 | 18.74 | 21.88 | 26.84 | 18.05 | 19.06 | 27.29 | 18.55 | 21.59 |
| *Backbone: Llama-3-8B* | | | | | | | | | |
| SLEB (Llama) | 13.00 | 13.00 | 13.00 | 13.00 | 13.00 | 13.00 | 13.00 | 13.00 | 13.00 |
| AdaInfer (Llama) | 4.00 | 0.00 | 3.18 | 3.00 | 1.00 | 4.71 | 0.00 | 2.00 | 2.24 |
| *RAEE* (Llama) | 11.77 | 15.70 | 12.43 | 7.04 | 12.83 | 6.58 | 20.06 | 21.04 | 13.43 |
| *Backbone: Gemma-7B* | | | | | | | | | |
| SLEB (Gemma) | 11.00 | 11.00 | 11.00 | 11.00 | 11.00 | 11.00 | 11.00 | 11.00 | 11.00 |
| AdaInfer (Gemma) | 1.00 | 0.00 | 1.04 | 1.00 | 3.00 | 1.00 | 0.00 | 2.00 | 1.13 |
| *RAEE* (Gemma) | 11.00 | 17.62 | 11.70 | 3.29 | 14.72 | 0.51 | 9.50 | 20.06 | 11.05 |

- For DeeBERT(Xin et al., 2020), we use RoBERTa-Large as its backbone model. Since DeeBERT(Xin et al., 2020) is a classical entropy-thresholding-based early-exit method, it requires first fine-tuning the backbone model on the downstream task and then updating all but the last off-ramp, for a fair comparison, we only update the off-ramp in DeeBERT on each downstream task. We also use RoBERTa-large as the backbone model and train all off-ramps for 50 epochs (much larger than the default setting of 10 epochs). Other experimental settings for DeeBERT(Xin et al., 2020) remain as default.

- For CALM (Schuster et al., 2022), we use T5-Large (Raffel et al., 2020) as its backbone model. CALM (Schuster et al., 2022) is also a classical entropy-thresholding-based early-exit method, and we evaluate it under the training-free setting.

- For SLEB(Song et al., 2024), we use Llama-3-8b (Dubey et al., 2024) and Gemma-7B (Mesnard et al., 2024) as its backbone model. SLEB(Song et al., 2024) tackles the limitation of early exit methods by eliminating redundant transformer blocks. Since the proposed RAEE exits at about 40% layers, for a fair comparison, we also set the hyper-parameter `num_remove_blocks` of SLEB(Song et al., 2024) as $\text{int}(60\% \cdot \text{num\_layers})$ for comparable efficiency.

## F  EXPERIMENTAL SETUP FOR OUT-OF-DOMAIN PERFORMANCE

This section details the experimental setup for the out-of-domain performance analysis.

To investigate out-of-domain issues, we constructed a retrieval database based on WikiText-2-v1 (Merity et al., 2017). Since the text dataset lacks gold labels, we adopted the next-token prediction task setting, where the next token of each input sentence serves as the gold label.

Specifically, we first segmented the entire text dataset into individual sentences to preserve semantic integrity. Then, based on the maximum input length of the backbone model, for sentences shorter than this limit, we designated the last meaningful token as the gold label. For longer sentences, we applied a sliding window of the model's maximum input length, selecting the last meaningful token within each window as the gold label. Finally, we collected exit information using the approach described in this paper.

Using the retrieval database constructed from WikiText (Merity et al., 2017), we conducted summarization experiments on CNN/DailyMail (See et al., 2017) and XSum (Narayan et al., 2018). The evaluation metrics for these tasks align with those used for summarization tasks in LongBench (Bai et al., 2024)with the zero-shot setting.

## G  RETRIEVED EXAMPLES OF RAEE

We show two examples from the SST-2 task and their retrieved top-k data samples. As shown in Table 10 and Table 11, the retrieved samples are semantically similar to the query sentence, demonstrating the proposed RAEE's efficacy.

## H  LAYER-WISE EARLY-EXIT DISTRIBUTIONS

To show a visualization of early-exit behavior, we report the proportion of samples that exit at each layer for eight downstream tasks. Figures 4 and 5 show the layer-wise early-exit distributions of RAEE when applied to **RoBERTa-large** and **LLaMA-3-8B**, respectively. For each task, the $x$-axis denotes the exit layer index and the $y$-axis denotes the fraction of test samples that terminate at that layer. These figures present results consistent with the latency trends in Figure 3, providing more fine-grained evidence of how RAEE redistributes computation across layers. For example, RAEE exits mostly at the 1st and 11th layers in Figure 5(g), leading to a substantial speedup as shown in Figure 3(b). In contrast, RAEE exits predominantly around the 27th layer in Figure 5(h), which results in a smaller speedup in Figure 3(c) when using the LLaMA-3-8B backbone. For the RoBERTa-large backbone, the exit distributions in Figure 4(a) and Figure 4(g) are similar, which is consistent with the comparable inference latencies reported in Figures 3(b) and Figure 3(d).

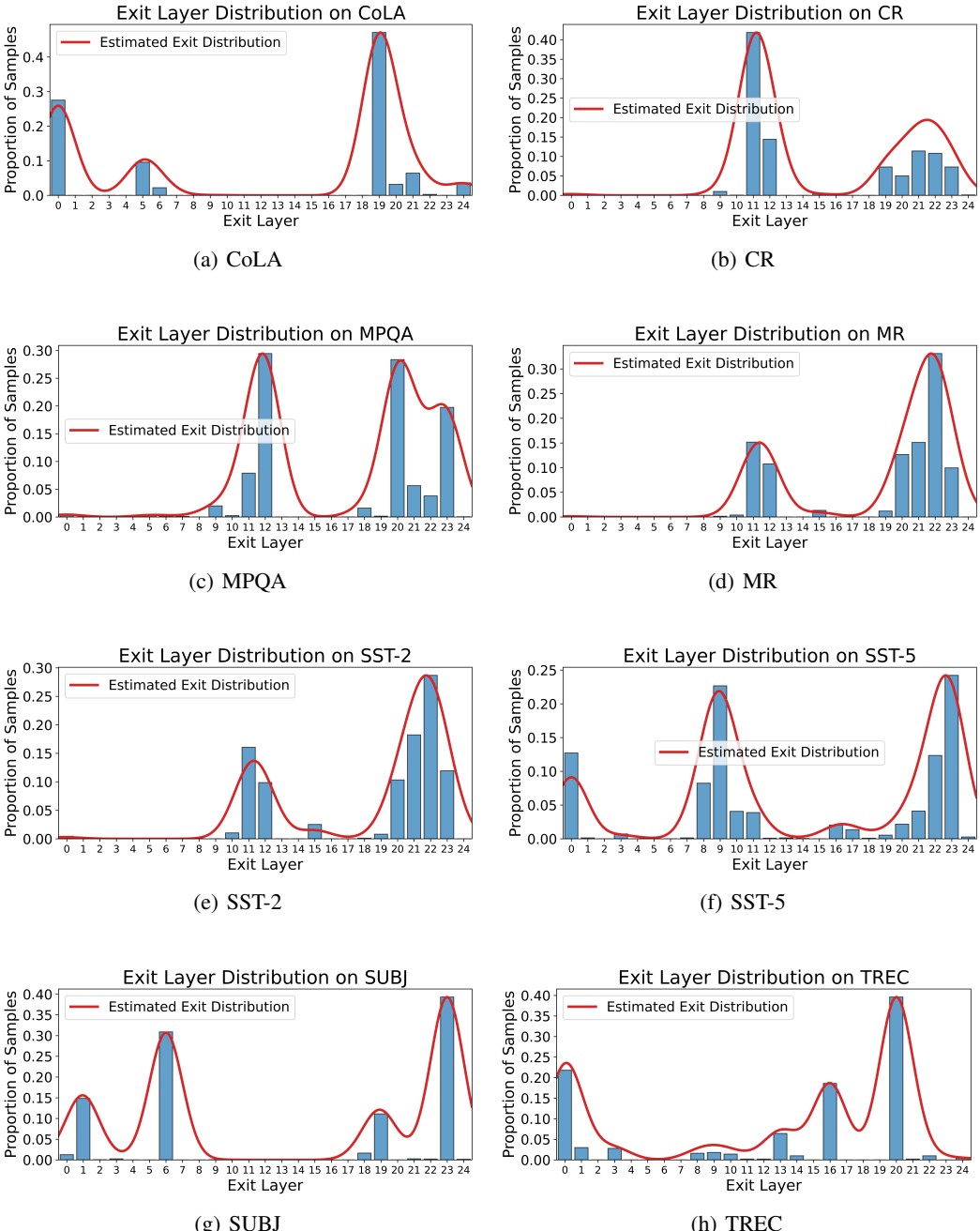

Figure 4: Layer-wise early-exit distributions of RAEE with a **RoBERTa-large** backbone on eight downstream tasks. For each task, the $x$-axis is the exit layer and the $y$-axis is the proportion of samples that terminate at that layer.

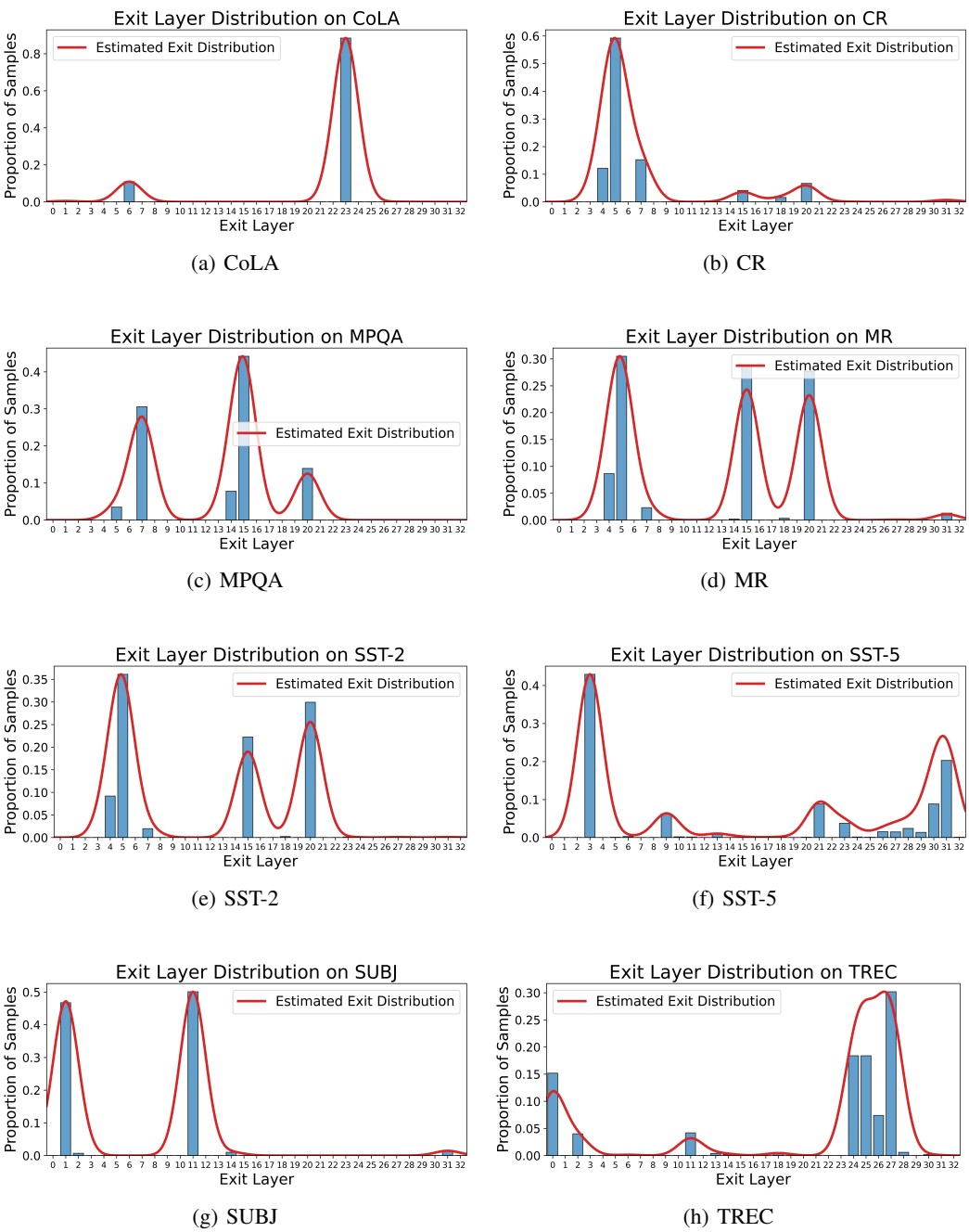

Figure 5: Layer-wise early-exit distributions of RAEE with a **LLaMA-3-8B** backbone on eight downstream tasks. For each task, the $x$-axis is the exit layer and the $y$-axis is the proportion of samples that terminate at that layer.

Table 10: Examples of data and corresponding retrieved data.

| Query/Top-K | Sentence | Label |
|---|---|---|
| Query | although laced with humor and a few fanciful touches, the film is a refreshingly serious look at young women. | 1 |
| Top-1 | the film is hard to dismiss – moody, thoughtful, and lit by flashes of mordant humor. | 1 |
| Top-2 | the movie enters a realm where few non-porn films venture, and comes across as darkly funny, energetic, and surprisingly gentle. | 1 |
| Top-3 | the movie, despite its rough edges and a tendency to sag in certain places, is wry and engrossing. | 1 |
| Top-4 | metaphors abound, but it is easy to take this film at face value and enjoy its slightly humorous and tender story. | 1 |
| Top-5 | it may not be particularly innovative, but the film's crisp, unaffected style and air of gentle longing make it unexpectedly rewarding. | 1 |
| Top-6 | it has its faults, but it is a kind, unapologetic, sweetheart of a movie, and mandy moore leaves a positive impression. | 1 |
| Top-7 | although frailty fits into a classic genre, in its script and execution it is a remarkably original work. | 1 |
| Top-8 | unlike lots of hollywood fluff, this has layered, well-developed characters and some surprises. | 1 |
| Top-9 | as broad and cartoonish as the screenplay is, there is an accuracy of observation in the work of the director, frank novak, that keeps the film grounded in an undeniable social realism. | 1 |
| Top-10 | though its rather routine script is loaded with familiar situations, the movie has a cinematic fluidity and sense of intelligence that makes it work more than it probably should. | 1 |
| Top-11 | it tends to remind one of a really solid woody allen film, with its excellent use of new york locales and sharp writing. | 1 |
| Top-12 | though a touch too arthouse 101 in its poetic symbolism, heaven proves to be a good match of the sensibilities of two directors. | 1 |

Table 11: Examples of data and corresponding retrieved data (Cond).

| Query/Top-K | Sentence | Label |
|---|---|---|
| Query | ... a boring parade of talking heads and technical gibberish that will do little to advance the linux cause. | 0 |
| Top-1 | a vile, incoherent mess... a scummy ripoff of david cronenberg's brilliant 'videodrome. | 0 |
| Top-2 | completely creatively stillborn and executed in a manner that i'm not sure could be a single iota worse... a soulless hunk of exploitative garbage. | 0 |
| Top-3 | contrived, maudlin and cliche-ridden... if this sappy script was the best the contest received, those rejected must have been astronomically bad. | 0 |
| Top-4 | could as easily have been called ' under siege 3: in alcatraz '... a cinematic corpse that never springs to life. | 0 |
| Top-5 | little more than a stylish exercise in revisionism whose point...is no doubt true, but serves as a rather thin moral to such a knowing fable. | 0 |
| Top-6 | a thoroughly awful movie – dumb, narratively chaotic, visually sloppy...a weird amalgam of 'the thing' and a geriatric scream. | 0 |
| Top-7 | on a cutting room floor somewhere lies...footage that might have made no such thing a trenchant, ironic cultural satire instead of a frustrating misfire. | 0 |
| Top-8 | ...while certainly clever in spots, this too-long, spoofy update of shakespeare's macbeth does n't sustain a high enough level of invention. | 0 |
| Top-9 | worthless, from its pseudo-rock-video opening to the idiocy of its last frames. | 0 |
| Top-10 | comes across as a relic from a bygone era, and its convolutions... feel silly rather than plausible. | 0 |
| Top-11 | a tired, unnecessary retread...a stale copy of a picture that was n't all that great to begin with. | 0 |
| Top-12 | (less a movie than) an appalling, odoriferous thing...so rotten in almost every single facet of production that you'll want to crawl up your own in embarrassment. | 0 |

