# OpenReview forum: "RAEE: A Robust Retrieval-Augmented Early Exit Framework for Efficient Inference"
_ICLR.cc/2026/Conference — ICLR 2026 Poster_

### Official Review · Reviewer_jtM8 · 2025-10-20

**Soundness:** 3
**Presentation:** 3
**Contribution:** 3
**Rating:** 6
**Confidence:** 4

**Summary:**

This paper proposes RAEE, a robust Retrieval-Augmented Early Exiting framework for efficient inference. RAEE first builds the retrieval dataset that records the correct exit layer, then retrieves the nearest neighbor to get the exit layer. Experimental results show its effectiveness.

**Strengths:**

1. Using $k$NN-based retrieval method for early exiting is novel.
2. This paper is simple, effective, and intuitive, even without additional technical contributions.
3. Experimental results are good.

**Weaknesses:**

1. The paper lacks a visualization of early-exit layers. What proportion of data exits early at each layer for each dataset?

2. Why does your method introduce additional time overhead on T5 in Figure 3?

3. Is the proposed method effective for supervised methods? For example, for a T5 model fine-tuned on SST-2.

4. Section 4.5 may not be very solid. Could you provide the performance when using a fixed layer, for example, the six values from layers 27 to 32?

**Questions:**

See weakness.

---

> ### Comment · Reviewer_jtM8 · 2025-11-25
>
> I am not sure why the authors did not submit a rebuttal. From my perspective, the paper’s initial score was already highly competitive, and I also really appreciate its simple yet effective approach. I adjust my score to a 4.

---

> > ### Author Response · Authors · 2025-11-26
> > **Comment:**
> >
> > We are sorry for not replying in time due to severe computing resource constraints during the rebuttal period. We sincerely appreciate that you found our approach simple yet effective, and we are very grateful for your insighful suggestions. The followings are the detailed responses to your comments.
> >
> > ### W1. Visualization of early-exit layers.
> >
> > R: Thanks for your suggestions. We visualized the early-exit layers of RAEE on RoBERTa-large and Llama-3-8B in the Figure 4 and Figure 5 in Appendix H. We analyzed the early exit patterns by plotting a histogram of the exit layers for each data sample, followed by fitting a Kernel Density Estimation (KDE) curve to visualize the underlying distribution. These figures present the results consistent with Figure 3 in the manuscript, providing more detailed evidence of the speed difference between RAEE and the baseline. For example, RAEE exits mostly on the 1st layer and the 11th layer in Figure 5 (g), delivering a significant speedup as shown in Figure 3 (b).
> >
> > ### W2. Additional time overhead on T5.
> >
> > R: Thanks for your comments. The RAEE’s inference latency is comprised of two parts: (1) the retrieving time including encoding time and ANN searching time; (2) early-exit-based forwarding time. Due to the plotting issue of using unified y scale, the results may be not significant in Figure 3. Specifically, the average inference latency of RAEE on T5-large over 8 tasks is about 6.63 ms on average (including 1.80 for encoding, 0.56 for ANN searching, and 4.27 for forwarding), while the original T5-large's inference latency is about 6.20 on average. Although RAEE's total inference latency is a bit higher than that of the original T5-large, the model forwarding latency is significantly reduced. The major overhead comes from the encoding part as we adopt a BERT-base model as our encoder, which has the same magnitude of parameters (million-level) compared to T5-large. This overhead can be further reduced by using more lightweight encoding method at the cost of less accurate semantic representations. However, when applying RAEE to existing LLMs, this overhead is trivial and can be almost negligible.
> >
> > ### W3. Application to supervised methods.
> >
> > R: Thanks for your comments. We apply our RAEE on a fine-tuned RoBERTa-large model (the public checkpoint philschmid/roberta-large-sst2) which is trained on the SST-2 data. The results in Rebuttal-Table 1 show that our RAEE can not only maintain a comparable inference efficiency, but also further improve the accuracy from 96.22\% to 96.33\%. The results demonstrate that RAEE is also effective when the backbone model is fully fine-tuned on the downstream task.  Later, we will try to apply our RAEE to more fine-tuned models on more tasks.
> >
> > Rebuttal-Table 1.  RAEE on a supervised RoBERTa-large model fine-tuned on SST-2 (philschmid/roberta-large-sst2).
> > | Model                              | Acc. (%) | Avg Exit Layer |
> > |------------------------------------|----------|------------------|
> > | RoBERTa-Large-SST2 (Supervised)    | 96.22    | 24.00            |
> > | RAEE + RoBERTa-Large-SST2          | 96.33    | 23.86            |

---

> > ### Author Response · Authors · 2025-11-26
> > **Comment:**
> >
> > ### W4. OOD Performance using a fixed-layer early-exit strategy.
> > R: Thanks for your comments. We follows your instructions to evaluate the out-of-domain performance using a fixed-layer early-exit strategy (from 24 to 32). The results on CNN/DailyMail and XSum in Rebuttal-Table 2 show that although the full model doesn't perfom well, it achieves a better performance when exiting from layer 24 to 31. This offers the opportunity to correct the model using early-exit. Therefore, compared to the fixed-layer early-exit strategy, RAEE dynamically exits based on the inputs, thus makes a better trade-off between efficency and accuracy. More importantly, although exiting on some fixed layers can achieve a better Rouge-L, it is hard to determine the fixed layer to exit on new models and new datasets in advance.
> >
> > Rebuttal-Table 2. OOD Performance of RAEE with fixed exit layers on CNN/DailyMail and XSum using the backbone model LLaMA-3-8B.
> > | Model                       | Avg Exit Layer (CNN/DM) | ROUGE-L (CNN/DM) | Avg Exit Layer (XSum) | ROUGE-L (XSum) |
> > |-----------------------------|--------------------------|------------------|------------------------|----------------|
> > | Llama                       | 32.00                    | 8.95             | 32.00                  | 5.22           |
> > | RAEE Wiki                   | 29.60                    | 14.01            | 28.82                  | 7.15           |
> > | RAEE–Fixed-L31              | 31.00                    | 22.16            | 31.00                  | 12.52          |
> > | RAEE–Fixed-L30              | 30.00                    | 21.52            | 30.00                  | 10.63          |
> > | RAEE–Fixed-L29              | 29.00                    | 18.87            | 29.00                  | 9.28           |
> > | RAEE–Fixed-L28              | 28.00                    | 17.72            | 28.00                  | 8.81           |
> > | RAEE–Fixed-L27              | 27.00                    | 13.59            | 27.00                  | 8.09           |
> > | RAEE–Fixed-L26              | 26.00                    | 10.91            | 26.00                  | 5.90           |
> > | RAEE–Fixed-L25              | 25.00                    | 10.71            | 25.00                  | 6.90           |
> > | RAEE–Fixed-L24              | 24.00                    | 9.05             | 24.00                  | 5.29           |

---

> > > ### Comment · Reviewer_jtM8 · 2025-11-26
> > >
> > > Thanks for your reply. I have adjusted the score back to 6 and will provide an evaluation later.

---

> > > ### Comment · Reviewer_jtM8 · 2025-11-27
> > >
> > > Thank you for your response. After reviewing the other reviewers’ comments, I will maintain my positive score and support the acceptance of this paper. I also agree that including more recent baseline methods would strengthen the paper. Good luck!

---

### Official Review · Reviewer_qNzk · 2025-10-31

**Soundness:** 3
**Presentation:** 3
**Contribution:** 4
**Rating:** 8
**Confidence:** 3

**Summary:**

This paper proposes RAEE (Retrieval-Augmented Early Exit), a training-free early exit framework that leverages a retrieval database built from the training set to dynamically select an optimal exit layer during inference. The key insight is that early exit can serve not only as an acceleration mechanism but also as a **corrective** one—intermediate layers sometimes outperform the final layer. RAEE models early exit as a distribution prediction problem and approximates this distribution by aggregating exit behaviors of semantically similar training examples retrieved via FAISS. Experiments across eight GLUE tasks and multiple backbone models (RoBERTa-Large, T5-Large, Llama-3-8B, Gemma-7B) show that RAEE consistently reduces latency (by nearly half for large models) while **improving accuracy**, often surpassing the full model baseline.

**Strengths:**

1. Originality: Proposes a novel perspective—early exit as a corrective mechanism—and implements it via retrieval augmentation, avoiding the need for trainable classifiers.
2. Quality: Comprehensive experiments across 8 tasks and 4 model families; includes ablation on $k$, database size, and OOD generalization.
3. Clarity: Figures 1–2 and Algorithms C.1–C.2 make the method transparent. The “correct ratio” analysis (Figure 1b) is particularly compelling.
4. Significance: Offers a practical, zero-shot solution to LLM inference efficiency that simultaneously boosts accuracy—a rare win-win in model compression.

**Weaknesses:**

1. Dependency on in-distribution labeled data: RAEE requires access to the **training set with ground-truth labels** to build the database. This limits applicability in zero-shot or unsupervised settings. The paper acknowledges this but does not explore alternatives (e.g., using pseudo-labels or confidence-based proxies).
2. Inconsistent gains across tasks: Performance improvements vary widely (e.g., CoLA: +12.45%; SST-2: +1.03% with RoBERTa). The paper does not analyze **why**—e.g., whether task difficulty, label distribution, or model calibration plays a role.
3. Embedding quality sensitivity: The method assumes that the query embedding (from backbone or BERT) captures semantic similarity well. No ablation compares different embedding sources (e.g., [CLS] vs. mean pooling vs. Sentence-BERT).
4. Limited OOD validation: Only tested on summarization with WikiText→CNN/XSum. More diverse OOD shifts (e.g., domain, style, language) would better assess robustness.
5. Scalability concerns: The retrieval component's latency with larger databases remains unexplored, raising practical deployment questions.

**Questions:**

1. The performance gain is dramatic on CoLA but marginal on SST-2. Can the authors provide an analysis of **which types of tasks or data distributions benefit most** from RAEE? Is it related to task difficulty or the full model’s calibration?

2. RAEE requires ground-truth labels to build the database. Have the authors considered a **self-supervised variant** that uses high-confidence predictions (e.g., $p > 0.95$) as pseudo-labels? This would extend applicability to unlabeled data.

3. In Algorithm C.1, the key is the embedding from the backbone’s first layer ($h_0$). How sensitive is performance to the choice of embedding layer or pooling strategy? A brief ablation would strengthen the method’s robustness claim.

4. The OOD experiment uses a next-token prediction setup on WikiText. Could RAEE be adapted to **classification tasks without labels** in the OOD setting? If not, what are the fundamental limitations?

5. Given the observed variance in baseline performance, could the authors provide more details about their baseline implementation and evaluation protocol? Specifically, were the reported baseline scores obtained through standard evaluation procedures or customized setups?

6. As the retrieval database scales up, how does the method maintain inference efficiency? Have the authors tested the framework with larger database sizes (e.g., 100K+ examples) and measured the corresponding latency impact?

---

> ### Author Response · Authors · 2025-12-01
> **Comment:**
>
> ### W1, Q2. Dependency on in-distribution labeled data.
>
> R: Thanks for your suggestions. We agree that existing RAEE may be limited by the labelled data, but this also limits those early-exit works that require training the classifiers on labelled data. To address this limitation, we also have already explored using unlabelled, raw text data (e.g., WikiText-2) to build the retrieval database in Section 4.5. Notably, this is only a simple application of RAEE on unlabelled data, similar to the next-token prediction task, worthing further exploration in the future. We evaluate RAEE on two representative generation tasks (e.g., CNN/DailyMail and XSum) based on the retrieval database built on the WikiText-2. Experimental results show the consistent improvements on model performance (e.g., from 8.95 to 14.01 on ROUGE-L on CNN/DailyMail). This shows the great potential of applying RAEE in unsupervised settings, although it can achieve a much better improvements on both performance and efficiency with a labelled datasets.
>
> For question 2, we agree that we can use a self-supervised variant to generate pseudo labels when it has a high confidence, which would greatly enhance RAEE's applicability.
> However, there is key challenge: the potential disconnect between confidence and correctness, which means the model can be highly confident in wrong predictions when exiting in early layers.
> For RAEE built with labelled data, exit information collected by wrong labels may pollute the retrieval database, thus cumulating errors and misguiding exit decisions.
> We would like to investigate this in our future work.
>
> ### W2, Q1. Inconsistent gains across tasks.
>
> R: Thanks for your suggestions. We agree that the improvements of RAEE over baselines are not uniform across tasks, and we classify the downstream tasks into three categories based on the improvements:
>
> (1) Modest on relatively “easy” or well-solved tasks (e.g., SST-2: +1.03; MR: +0.75); The data of these tasks typically have clear sentiment signals, where existing models have already achieved great performance. Although the correction space is samll, RAEE can further improve the accuracy a little bit.
>
> (2) Very large on harder / poorly calibrated tasks where the baseline is weak (e.g., SUBJ: +32.60; TREC: +30.00; CoLA: +12.45; MPQA: +10.95); These tasks, such as SUBJ (subjective vs. objective sentence classification with diverse domains), TREC (small-scale, 6-way question-type classification), CoLA (grammatical acceptability judgments, a notoriously low-MCC GLUE task), and MPQA (fine-grained opinion polarity with subtle contextual cues), are widely regarded as more challenging and calibration-sensitive. The backbone model perform poorly on these tasks, while RAEE can well capture the early exit layer where the model already make the correct predictions.
>
> (3) Small drops on two tasks (SST-5: −1.41, CR: −11.50). SST-5 is a fine-grained five-class sentiment task, which is known to be harder and less stable than binary sentiment (SST-2), while CR is a relatively small customer-review dataset where overfitting and label imbalance can make improvements less consistent. Even in these cases, RAEE still provides non-trivial speedups without incurring substantial performance loss.

---

> ### Author Response · Authors · 2025-12-01
> **Comment:**
>
> ### W3, Q3. Embedding quality sensitivity
>
> R: Thanks for your suggestions. The default embedding strategy in our RAEE is mean pooling on BERT-base-uncased using SentenceTransformer. To evaluate the impact of different embedding strategies, we conducted ab ablation study on RoBERTa-large over eight downstream tasks. The embedding strategies include (1) RAEE using the '[CLS]' token embedding of the BERT-base-uncased's last layer; (2) RAEE using the mean pooling embedding of the backbone's first/second/third layer; (3) RAEE using the '[CLS]' token embedding of the backbone's first/second/third layer; (4) RAEE using Sentence-BERT of bert-base-nli-mean-tokens; (5) RAEE using Sentence-BERT of all-mpnet-base-v2. We report the accuracy (MCC for CoLA) of each task and the average exiting layers in Rebuttal-Table 3 and Rebuttal-Table 4, respectively.
>
> The results show that all RAEE variants outperform the vanilla RoBERTa-large baseline. The mean-pooling configuration used in our manuscript achieves the highest performance (63.41), while the best Sentence-BERT variant (bert-base-nli-mean-tokens) is only slightly behind (62.55). In addition, when we vary the internal source of query embeddings within RoBERTa, using BERT’s last-layer [CLS] as the retriever input attains an average score of 61.61, and mean-pooling over the first, second, and third backbone layers reaches 59.82, 61.40, and 61.62, respectively—all clearly above the vanilla baseline. Using backbone [CLS] embeddings yields somewhat lower performance (56.05, 56.78 and 56.16), but still consistently improves over the baseline model. Importantly, the average exit depth remains stable (around 15–16 layers vs. 24 layers for the full model) for all embedding choices, indicating that RAEE’s computation savings are robust.
>
> For question 3, in Algorithm C.1, we claimed that we can also use the hidden states as the embedding for retrieving the nearest neighbors if the extra encoder is missing. As shown in Rebuttal-Table 3, the model performance of RAEE using different layer's embedding and different pooling strategies are comparable, which indicates that it is not sensitive to the choice of embedding layer. However, RAEE using the mean pooling always achieves a better performance than that using [CLS] token. Besides, they all perform poorly compared to RAEE using the extra encoder due to weak representation capabilities.
>
> Rebuttal-Table 3. RAEE with different query embedding sources on eight downstream tasks (RoBERTa-Large backbone). CoLA is reported in MCC, others in accuracy (%).
>
> | Method                                                    | SST-2 | SST-5 |  MR  |  CR  | MPQA | SUBJ | TREC | CoLA | Avg-All |
> |---------------------------------------------------        |:-----:|:-----:|:----:|:----:|:----:|:----:|:----:|:----:|:-------:|
> | RoBERTa-Large                                             | 83.60 | 34.98 | 80.80 | 79.55 | 67.60 | 51.45 | 32.40 |  2.03 |  54.05 |
> | **RAEE (RoBERTa-Large, BERT mean pooling, ours)**         | 84.63 | 33.57 | 81.55 | 68.05 | 78.55 | 84.05 | 62.40 | 14.48 | **63.41** |
> | RAEE (RoBERTa-Large, BERT **[CLS]**)                      | 82.11 | 31.81 | 81.10 | 67.75 | 77.30 | 83.15 | 63.00 | 6.68  | 61.61 |
> | RAEE – backbone **first-layer mean pooling**              | 83.26 | 30.81 | 78.95 | 69.70 | 76.45 | 77.50 | 58.60 | 3.27  | 59.82 |
> | RAEE – backbone **second-layer mean pooling**             | 82.45 | 30.45 | 79.90 | 71.50 | 76.10 | 78.40 | 64.20 | 8.19  | 61.40 |
> | RAEE – backbone **third-layer mean pooling**              | 83.72 | 32.40 | 79.40 | 71.40 | 75.65 | 79.85 | 66.40 | 4.13  | 61.62 |
> | RAEE – backbone **first-layer [CLS]**                     | 79.93 | 30.00 | 78.55 | 71.45 | 74.20 | 61.30 | 50.40 |  2.56 | 56.05 |
> | RAEE – backbone **second-layer [CLS]**                    | 80.62 | 29.95 | 78.55 | 69.45 | 74.65 | 70.35 | 52.00 | -1.36 | 56.78 |
> | RAEE – backbone **third-layer [CLS]**                     | 81.08 | 30.90 | 79.00 | 67.40 | 73.90 | 65.85 | 53.60 | -2.49 | 56.16 |
> | RAEE – **Sentence-BERT** (`bert-base-nli-mean-tokens`)    | 85.78 | 36.20 | 82.35 | 78.60 | 83.95 | 78.15 | 52.60 |  2.77 | 62.55 |
> | RAEE – **Sentence-BERT** (`all-mpnet-base-v2`)            | 85.09 | 34.98 | 82.90 | 67.70 | 81.35 | 82.20 | 52.60 |  1.67 | 61.06 |

---

> ### Author Response · Authors · 2025-12-01
> **Comment:**
>
> Rebuttal-Table 4. Average exiting layers of RAEE with different query embeddings (RoBERTa-Large backbone).
>
> | Method                                                    | SST-2 | SST-5 |  MR  |  CR  | MPQA | SUBJ | TREC | CoLA | Avg-All |
> |---------------------------------------------------        |:-----:|:-----:|:----:|:----:|:----:|:----:|:----:|:----:|:-------:|
> | RoBERTa-Large                                             | 24.00 | 24.00 | 24.00 | 24.00 | 24.00 | 24.00 | 24.00 | 24.00 | 24.00 |
> | **RAEE (RoBERTa-Large, BERT mean pooling, ours)**         | 18.55 | 13.93 | 18.71 | 15.35 | 17.20 | 13.59 | 12.82 | 12.48 | 15.33 |
> | RAEE (RoBERTa-Large, BERT **last-layer [CLS]**)           | 18.47 | 13.69 | 18.91 | 15.06 | 17.57 | 14.00 | 13.54 | 13.24 | 15.56 |
> | RAEE – backbone **first-layer mean pooling**              | 19.27 | 14.21 | 19.09 | 16.44 | 17.71 | 13.01 | 14.07 | 13.55 | 15.92 |
> | RAEE – backbone **second-layer mean pooling**             | 19.17 | 13.99 | 19.07 | 16.58 | 18.08 | 12.62 | 13.03 | 13.00 | 15.69 |
> | RAEE – backbone **third-layer mean pooling**              | 19.07 | 13.92 | 18.90 | 16.66 | 18.07 | 12.52 | 13.27 | 13.76 | 15.77 |
> | RAEE – backbone **first-layer [CLS]**                     | 19.35 | 14.25 | 19.35 | 17.18 | 18.40 | 11.52 | 14.68 | 13.48 | 16.03 |
> | RAEE – backbone **second-layer [CLS]**                    | 19.30 | 14.23 | 19.31 | 16.52 | 18.56 | 11.73 | 14.78 | 12.92 | 15.92 |
> | RAEE – backbone **third-layer [CLS]**                     | 19.76 | 14.76 | 19.56 | 16.37 | 18.52 | 11.62 | 14.09 | 13.22 | 15.99 |
> | RAEE – **Sentence-BERT** (`bert-base-nli-mean-tokens`)    | 17.42 | 13.54 | 17.91 | 15.59 | 16.67 | 14.14 | 13.62 | 12.89 | 15.22 |
> | RAEE – **Sentence-BERT** (`all-mpnet-base-v2`)            | 17.21 | 14.42 | 17.68 | 14.99 | 16.98 | 14.50 | 13.62 | 13.90 | 15.41 |
>
> ### W4. Limited OOD validation.
>
> R: Thanks for your suggestions. We conducted two additional OOD experiments: (1) a new generation dataset with a different discourse style, and (2) cross-task OOD retrieval databases for zero-shot classification.
>
> (1) WikiText-based RAEE on DialogSum (generation, domain/style shift).
> We use the same WikiText-based retrieval database, built in the OOD experiments in Section 4.5, but evaluate our RAEE on DialogSum, a dialogue summarization benchmark whose conversational style differs substantially from encyclopedic WikiText and news summarization.
> As shown in Rebuttal-Table 5, RAEE increases ROUGE-L by +1.62 and skips about 1.3 layers on average, indicating that the method remains effective under domain and style shifts in generation.
>
> Rebuttal-Table 5. RAEE with a WikiText-based retrieval database evaluated on DialogSum (LLaMA-3-8B backbone).
>
> | Method                               | ROUGE-L | Avg Layers |
> |--------------------------------------|:-------:|:----------:|
> | DialogSum Llama-3-8B     |  11.69  |   32.00    |
> | DialogSum RAEE (Llama-wiki) |  **13.31** | **30.67**  |
>
> (2) Cross-task OOD retrieval databases on eight GLUE downstream tasks (classification).
> To systematically study OOD robustness for classification, we conducted a second experiment where we decouple the training corpus used to build the RAEE retrieval database from the target evaluation task. We use RoBERTa-large as a backbone and consider eight tasks. For each source task, we build a retrieval database using the training split of the task and evaluate RAEE in zero-shot prompt mode on all eight target tasks using that fixed database.
>
> The resulting accuracies (CoLA in MCC) are summarized in Rebuttal-Table 6, and the corresponding average exit layers are in Rebuttal-Table 7. And we can find that: OOD databases can still help.  For multiple targets, the best or competitive performance is obtained with out-of-domain databases. For example, SST-2 and SST-5 benefit strongly from databases built on other sentiment-related tasks (e.g., MR, MPQA), and MR/CR also obtain strong accuracy with database sources beyond themselves. This suggests that RAEE does not strictly require perfectly in-distribution retrieval data to be effective.

---

> ### Author Response · Authors · 2025-12-01
> **Comment:**
>
> Rebuttal-Table 6. Zero-shot RAEE accuracy with retrieval databases built from different source tasks (RoBERTa-large backbone). Diagonal entries (e.g., DB from SST-2 evaluated on SST-2) correspond to in-domain databases; off-diagonals are OOD. CoLA is reported in MCC, others in accuracy (%).
>
> | Retrieval DB (source task) | SST-2 | SST-5 |   MR  |   CR  | MPQA | SUBJ | TREC | CoLA |
> |---------------------------|:-----:|:-----:|:-----:|:-----:|:----:|:----:|:----:|:----:|
> | DB from **SST-2**         | 84.63 | 34.39 | 84.80 | 80.75 | 59.60 | 50.65 | 28.20 |  7.86 |
> | DB from **SST-5**         | 75.34 | 33.57 | 73.85 | 69.20 | 62.85 | 50.35 | 18.60 |  3.22 |
> | DB from **MR**            | 90.48 | 36.11 | 81.55 | 81.80 | 66.05 | 51.10 | 24.60 | -3.83 |
> | DB from **CR**            | 74.66 | 32.67 | 75.40 | 68.05 | 55.60 | 49.00 | 16.60 |  2.58 |
> | DB from **MPQA**          | 86.12 | 34.16 | 83.30 | 83.50 | 78.55 | 50.00 | 29.40 | -1.56 |
> | DB from **Subj**          | 79.70 | 32.22 | 75.90 | 72.55 | 54.70 | 84.05 | 11.80 |  8.06 |
> | DB from **TREC**          | 61.01 | 26.97 | 56.55 | 75.50 | 56.30 | 49.65 | 62.40 | -1.39 |
> | DB from **CoLA**          | 76.72 | 33.85 | 73.95 | 68.05 | 53.25 | 57.20 | 18.00 | 14.48 |
>
> Rebuttal-Table 7. Average exiting layers (smaller is faster) of RAEE with cross-task retrieval databases on the same eight tasks (RoBERTa-large backbone).
>
> | Retrieval DB (source task) | SST-2 | SST-5 |   MR  |   CR  | MPQA | SUBJ | TREC | CoLA |
> |---------------------------|:-----:|:-----:|:-----:|:-----:|:----:|:----:|:----:|:----:|
> | DB from **SST-2**         | 18.55 | 18.57 | 18.60 | 19.09 | 17.90 | 17.95 | 19.41 | 18.56 |
> | DB from **SST-5**         | 14.23 | 13.93 | 13.67 | 14.09 | 18.03 | 12.37 | 15.34 | 15.47 |
> | DB from **MR**            | 18.52 | 19.06 | 18.71 | 19.19 | 17.39 | 18.20 | 18.25 | 19.39 |
> | DB from **CR**            | 17.10 | 17.24 | 17.59 | 15.35 | 13.18 | 18.36 | 18.38 | 16.32 |
> | DB from **MPQA**          | 21.71 | 21.64 | 21.66 | 21.45 | 17.20 | 22.09 | 21.79 | 20.34 |
> | DB from **Subj**          | 18.48 | 18.02 | 18.06 | 16.75 |  7.67 | 13.59 |  6.94 | 11.09 |
> | DB from **TREC**          | 12.29 | 12.39 | 11.74 | 16.37 | 13.60 | 12.04 | 12.82 | 16.12 |
> | DB from **CoLA**          | 16.36 | 15.73 | 16.00 | 14.47 |  5.28 | 14.72 |  8.35 | 12.48 |
>
> ### W5, Q6. Scalability on larger databases.
>
> R: Thanks for your suggestions. To evaluate the scability of RAEE, we conducted a separate experiment on the efficiency of the retriever. We use ANN_GIST1M which consists of one million 960-dim GIST descriptor embeddings to build the retrieval database. We use the same configureation of retriever indexing (i.e., IVF665_HNSW32, PQ64 for index, 512 for probing, 12 for top-$k$, 32 for retrieve batch size). We report the average retrieving latency varying the database size from 10K to 1M. The results are summarized in Rebuttal-Table 8 below. As shown in Rebuttal-Table 8, even when scaling the database to 1M high-dimensional (960-D) vectors, the average retrieval latency remains around 0.77 ms per query. Since RAEE uses 768-D query embeddings in our actual downstream experiments, the practical retrieval cost is even lower in our main setups. This confirms that the retrieval component is not a bottleneck in realistic deployments, even under large-scale database sizes.
>
> Rebuttal-Table 8. Latency of FAISS retrieval on TexMex ANN_GIST1M with RAEE-style index settings.
>
> | #DB vectors N | Total search time (s) | Avg latency (ms / query) |
> | ------------- | --------------------- | ------------------------ |
> | 10,000        | 0.1725                | 0.172                    |
> | 50,000        | 0.3172                | 0.317                    |
> | 100,000       | 0.3563                | 0.356                    |
> | 300,000       | 0.4256                | 0.426                    |
> | 1,000,000     | 0.7663                | 0.766                    |

---

> ### Author Response · Authors · 2025-12-01
> **Comment:**
>
> ### Q4. OOD experiments of RAEE using unlabelled data on classification tasks.
>
> R: Thank you for the suggestion. Our OOD experiments in Rebuttal-Table 9 and Rebuttal-Table 10 reveal a key limitation: RAEE fails on classification tasks when using a generic, unlabeled database like WikiText.
>
> The failure occurs because the required reasoning for classification is absent in the retrieval database. Tasks like sentiment analysis (SST-2, MR) and grammatical judgment (CoLA) depend on recognizing specific features (e.g., subjective opinions, grammatical errors). WikiText, composed of encyclopedic facts, lacks these features. Retrieving a "similar" sentence for a movie review yields factual descriptions, not other texts with sentiment flags or words. The exit behavior of these irrelevant examples provides no useful signal for the classification task, leading to performance degradation and minimal speedup.
>
> In contrast, RAEE succeeds in generation tasks (e.g., summarization on CNN/DailyMail, XSum) with the same WikiText database. This is because the core capability required for summarization, understanding and condensing general language, is directly supported by the WikiText corpus, allowing its exit behavior to transfer effectively.
>
> These results demonstrate that RAEE's effectiveness in OOD settings depends critically on the semantic and functional alignment between the retrieval database and the downstream task's core reasoning requirements.
>
> Rebuttal-Table 9. RAEE on eight classification tasks with LLaMA-3-8B as backbone. CoLA is reported in MCC, others in accuracy (%). “RAEE (Llama-'Wiki') ” uses an OOD retrieval database built on WikiText-2 without task labels.
>
> | Method                             | SST-2 | SST-5 |  MR   |  CR   | MPQA | SUBJ | TREC | CoLA | Avg-All |
> |------------------------------------|:-----:|:-----:|:-----:|:-----:|:----:|:----:|:----:|:----:|:-------:|
> | LLaMA-3-8B   | 62.84 | 26.06 | 59.65 | 72.90 | 51.75 | 52.80 |  8.40 | 0.00 |  41.80 |
> | RAEE (Llama)     | 73.05 | 35.25 | 66.45 | 57.95 | 75.05 | 90.05 | 51.80 | 9.55 |  57.39 |
> | RAEE (Llama-'Wiki')  | 55.50 | 21.40 | 54.30 | 61.60 | 57.15 | 51.55 | 13.00 | 0.00 |  39.31 |
>
> Rebuttal-Table 10. Average exit layers (smaller is faster) for the same setting as Rebuttal-Table 9.
>
> | Method                             | SST-2 | SST-5 |  MR   |  CR   | MPQA | SUBJ | TREC | CoLA | Avg-All |
> |------------------------------------|:-----:|:-----:|:-----:|:-----:|:----:|:----:|:----:|:----:|:-------:|
> | LLaMA-3-8B   | 32.00 | 32.00 | 32.00 | 32.00 | 32.00 | 32.00 | 32.00 | 32.00 | 32.00  |
> | RAEE (Llama)     | 11.77 | 15.70 | 12.43 |  7.04 | 12.83 |  6.58 | 20.06 | 21.04 | 13.43  |
> | RAEE (Llama-'Wiki')  | 29.41 | 28.83 | 29.30 | 28.75 | 30.13 | 29.31 | 30.34 | 30.56 | 29.58  |
>
> ### Q5. More details about baseline implementation and evaluation protocol.
>
> R: Thanks for your comments regarding baseline implementation and evaluation protocol. We confirm that all baselines were implemented and evaluated under a standardized, fair protocol to ensure a valid comparison.
>
> The key implementation details for the baselines are as follows:
> * HashEE: We used the official codes and used the pre-trained ElasticBERT-Large model without fine-tuning its parameters. The hashing procedure was run with its default parameters as specified in the original paper.
> * Adalnfer: Since it has no open-source codes, we implement it by ourself. We collect the features as described in its paper and we did not tune the backbone model's parameters for fair comparison. We train the SVM classifiers based on the features.
> * The detailed implementation for DeeBERT, CALM, and SLEB is already provided in Appendix E.
>
> All baselines (including the above, the full models, and our RAEE) were evaluated using the identical set of prompt templates from LM-BFF [1] to ensure a consistent prompting strategy across all methods. The prompt templates used for each task are listed in Tables 7 and 8 in Appendix B.
>
> [1] Tianyu Gao, Adam Fisch, Danqi Chen. Making Pre-trained Language Models Better Few-shot Learners. ACL/IJCNLP 2021.

---

### Official Review · Reviewer_niT3 · 2025-10-31

**Soundness:** 3
**Presentation:** 2
**Contribution:** 2
**Rating:** 2
**Confidence:** 5

**Summary:**

The paper presents a retrieval-augmented early exit framework that enhances the overall model performance by classifying the input at one of the intermediate exit points. The proposed RAEE model can outperform the full model performance while improving the inference speed of the model. The method does not require any training or additional classifiers to which makes the method efficient. The method is tested upon eight downstream tasks which further validates the method importance.

**Strengths:**

1. The method improves the efficiency of a model by early exiting where the decision to make an exit is based on the analysis of similar previously seen samples helping in training free-exiting.

2. The method utilises the data characteristics like word embedding to map the incoming sample to the existing database and then directly assigning the layer to exit based on the samples in database to which the incoming sample closely resembles.

**Weaknesses:**

1. While authors claim that the method is zero-shot while they use the training samples to create the database making the method supervised, this claim makes the proposed method confusing.

2. The provided method cannot be easily generalised to other tasks except classification which is a major issue with this work.

3. Why there are different baselines with different backbone, this makes it unfair for a fair comparison all methods should be tested on the same underlying backbone model. For instance, DeeBERT can be applied to T5 and Gemma but authors do not report.

4. The methodology of the work closely aligns with an existing work [1] still no comparison with it is found.

5. There are multiple methods which prove and show that if monitored properly early exits can outperform the full models which are not compared and not even cited which shows that the method has a less hold of the existing literature [2], [3], [4], [5] and many more.

[1] https://ieeexplore.ieee.org/document/11161749

[2] https://arxiv.org/abs/2502.00745 [ICLR 2025]

[3] https://proceedings.neurips.cc/paper_files/paper/2024/file/ea5a63f7ddb82e58623693fd1f4933f7-Paper-Conference.pdf

[4] https://openaccess.thecvf.com/content/WACV2024/papers/Meronen_Fixing_Overconfidence_in_Dynamic_Neural_Networks_WACV_2024_paper.pdf

[5] https://aclanthology.org/2025.findings-acl.1209.pdf

**Questions:**

See weaknesses.

---

> ### Author Response · Authors · 2025-12-03
> **Comment:**
>
> ### W1. Confusing claims of zero-shot setting.
>
> R: Thank you for raising this point regarding terminology.  We appreciate the opportunity to clarify.
>
> In the revised manuscript, we have replaced “zero-shot” with “training-free” to more accurately describe RAEE.  The term is intended to emphasize that **NO parameter updates or gradient-based optimization** are performed on the backbone model for the downstream task, *unlike training-based or semi-training early-exit methods that require updating classifiers or model weights.*
>
> While RAEE does use task data to build a retrieval database of exit behaviors, this process only collects successful exit patterns through inference and stores them statically.  **No backpropagation, loss minimization, or weight adjustment occurs.** At test time, predictions are made without including example data in the input (only a fixed prompt is used), and the retrieved exit information functions similarly to retrieving prior “expert decisions” to guide early exiting—closer in spirit to retrieval-augmented inference than to supervised training.
>
> We believe this clearly differentiates RAEE from methods that rely on learned parametric classifiers, and the revised terminology should prevent any further confusion.
>
> ### W2. Generalization to other tasks except classification.
>
> R: Thanks for your comments.  While most of our main experiments focus on the GLUE benchmark, RAEE is not limited to this type of tasks.  We have presented more experimental results in the manuscript and in this rebuttal, including RAEE on **generation tasks or on two reasoning tasks**.
>
> For generation tasks, we have leveraged WikiText to build the retrieval database and evaluated our RAEE on CNN/DailyMail (Table 5 in the manuscript), XSum (Table 5 in the manuscript), and DialogSum (Rebuttal-Table 5 in the responses to W5 of reviewer qNzk). For reasoning tasks, we added new two tasks, i.e., PIQA, which evaluates physical commonsense reasoning and ARC-Easy, a grade-school science question answering task, in Rebuttal-Table 11. All results show that RAEE can not only improve the end-to-end efficiency, but also further improve the model performance. These results indicate that RAEE can be effectively applied beyond standard classification benchmarks.
>
> Rebuttal-Table 11. RAEE on PIQA and ARC-Easy with the backbone model LLaMA-3-8B, reporting accuracy, average exit layers, and per-query inference latency (ms).
> | Task     | Model                          | Accuracy (%) | Avg. Exit Layers | Latency (ms) |
> |----------|--------------------------------|--------------|------------------|--------------|
> | PIQA     | LLaMA-3-8B          | 54.00        | 32.00            | 248.56       |
> | PIQA     | RAEE (Llama)      | 59.00        | 12.14            | 114.19       |
> | ARC-Easy | LLaMA-3-8B         | 59.00        | 32.00            | 248.89       |
> | ARC-Easy | RAEE (Llama)     | 65.00        | 25.08            | 202.36       |
>
> ### W3. Different baselines with different backbone.
>
> R: Thanks for your comments. Since not every early exit frameworks compared in this work are suitable for all kinds of model architecture, including the BEEM you mentioned seems only support bert-based backbones, we have tried our best to compare RAEE to mainstream methods on each model architecture. Specifically, HashEE and DeeBERT are designed and released only for BERT-like encoder-only architectures; CALM is implemented only for T5-style encoder–decoder architectures; and SLEB is implemented only for decoder-only architectures. Their paper and public codebases do not provide evidence of variants for other architectures, and extending them would require non-trivial re-designs of their confidence heads and layer-wise routing mechanisms, which is beyond the scope of our work. And we have tried our best to implement AdaInfer over all kinds of backbone models whose design is less related to the architecture of backbone models. **More importantly, our RAEE can be adapted to any Transformer-based models, including encoder-only, encoder-decoder, and decoder-only model architectures.**

---

> ### Author Response · Authors · 2025-12-03
> **Comment:**
>
> ### W4. Potentially comparison.
>
> R: Thanks for your comments. ***The ICC 2025 version of [1] was published in the conference proceedings on 26 September 2025, whereas the submission deadline of ICLR 2026 is 24 September 2025, so we didn't compare this work in our paper.***
>
> More importantly, the problem and the methodologies in [1] and our RAEE are significantly  different, which demonstrates that they are **NOT** closely aligned.
>
> RAEE's main goal is to improve accuracy over the backbone model by **correcting it** with a pre-built retrieval database, which contains the sets of possible exit layers with associated correctness probabilities of train data, while reducing inference latency.
> This is a general algorithm improvement method and it is not optimized for actual deployment scenarios (e.g., distributed system).
>
> In contrast, [1] applies early-exit-based BERT to a three-layer distributed inference system ("mobile/edge/cloud"), the goal is to determine which layer of the device ("mobile/edge/cloud") each sample is computed on based on the difficulty of the sample.    This method reduces the overall inference cost while achieving **comparable accuracy**.
>
> The main novelty of RAEE is that we found **the backbone model can be corrected with a retrieval-based early exit method, where both performance and efficiency can be improved simultaneously.**
> However, [1] contributes a cost-aware device assignment and threshold optimization framework for distributed inference.
> The two works are, therefore, complementary rather than “methodologically closely aligned.”
>
> ### W5. More literature works.
>
> R: Thanks for your comments. We will add those works [2–5] in the revised version and will discuss them in the related works.
>
> [2] BEEM (ICLR 2025).
> The paper proposes an early exit strategy that treats each intermediate classifier as an "expert," weights and accumulates their softmax confidence (accumulation only occurs when adjacent layers predict the same value, otherwise the confidence is reset), and exits early when the accumulated confidence exceeds a threshold. This strategy significantly accelerates the inference speed of BERT-like encoder-only models while maintaining similar model performance to the full model. However, the `run_glue.py` code required by the `expert_infer_(al)bert.sh` script used in the inference part of this paper is not present in the open-source code repository (the open-source code is incomplete), so we could not obtain a fully controlled comparison.
>
> [3] Fast yet Safe: Early-Exiting with Risk Control (NeurIPS 2024).
> The paper proposes a theoretically guaranteed "risk-controlled exit strategy" based on existing dynamic models/early-exit models (MSDNet, Dynamic ViT, CALM, diffusion models, etc.), and provides a general implementation. In its code repository, the Language Modeling task is provided in a `calm` folder. However, this folder only contains the pre-inferred CALM results in the pre-packaged `calm_cnndm.p/calm_squad.p` and the code for directly processing the CALM results. It does not provide the code for collecting these pre-packaged experimental results for CALM tasks. However, we have already conducted experiments comparing our method directly with the CALM method itself in the main experiments of our paper and provided the experimental results.
>
> [4] Fixing Overconfidence in Dynamic Neural Networks (WACV 2024).
> The paper mainly studies the overconfidence of dynamic early-exit CNNs such as MSDNet and proposes a cheap Bayesian post-hoc calibration scheme to make exit confidences more reliable, thereby improving the decision of whether to continue computing deeper layers. Their experiments are entirely on CNN-based visual classification models and image classification datasets, which are orthogonal to our LLM-centric on NLP tasks.
>
> [5] FREE: Fast and Robust Vision Language Models with Early Exits (Findings of ACL 2025).
> The paper proposes an adversarial, GAN-style training strategy to equip vision–language models (e.g., BLIP-2-style VLMs) with reliable early exits: each exit contains a transformer layer plus a classifier; the intermediate transformer layers are adversarially trained to mimic the final-layer representation, while a discriminator encourages alignment, so that early exits are both accurate and robust. This work is therefore about training new exit modules in multimodal VLMs, not about training-free exit policies for existing autoregressive LLMs. All tasks are performed using VLMs for multimodal tasks. Furthermore, the open-source code provided in this paper returns a 404 error (invalid open-source code link), which prevents us from running their method.

---

### Meta-Review · Area_Chair_oyQk · 2026-01-07

**Summary:**

This paper proposes RAEE (Retrieval-Augmented Early Exit), a training-free early exit framework that leverages a retrieval database built from the training set to dynamically select an optimal exit layer during inference. Experiments across eight GLUE tasks and multiple backbone models (RoBERTa-Large, T5-Large, Llama-3-8B, Gemma-7B) show that RAEE consistently reduces latency (by nearly half for large models) while improving accuracy, often surpassing the full model baseline.

While two of the reviewers evaluated this simple, intuitive but very effective method as a very novel research, one reviewer (niT3) concerns the over-claim of the research and also doubted the fair and sufficient comparison with recent similar research. I have carefully read the paper and rebuttals, and personally I would agree the positive comments. There are some overstatements but I think the authors have corrected them in the revised version, and the added experiments and literature reviews have addressed most of the negative evaluations. I do agree the paper provides a mechanism which consistently improves model performance and efficiency for different backbones, and I would tend to recommend this paper to the community.

**Reviewer Concerns:**

As mentioned above, the most concerns related to the completeness and robustness of the comparison have been addressed.

**Reviewer Scores:**

The original scores of the paper are 2(niT3), 8(qNzk), 6(jtM8). I would not expect there will be changes after the rebuttal discussion, however, I would still recommend the acceptance as from my perspective, the concerns of niT3 are not fatal to the value of this paper.

---

### Decision · Program_Chairs · 2026-01-26

Accept (Poster)